# SAFE POLICY LEARNING FROM OBSERVATIONS

## ABSTRACT

In this paper, we consider the problem of learning a policy by observing numerous non-expert agents. Our goal is to extract a policy that, with high-confidence, acts better than the agents' average performance. Such a setting is important for real-world problems where expert data is scarce but non-expert data can easily be obtained, e.g. by crowdsourcing. Our approach is to pose this problem as safe policy improvement in reinforcement learning. First, we evaluate an *average behavior* policy and approximate its value function. Then, we develop a stochastic policy improvement algorithm that safely improves the average behavior. The primary advantages of our approach, termed Rerouted Behavior Improvement (RBI), over other safe learning methods are its stability in the presence of value estimation errors and the elimination of a policy search process. We demonstrate these advantages in the Taxi grid-world domain and in four games from the Atari learning environment.

## 1   INTRODUCTION

Recent progress in Reinforcement Learning (RL) has shown a remarkable success in learning to play games such as Atari from raw sensory input (Mnih et al., 2015; Hessel et al., 2017). Still, tabula rasa, RL typically requires a significant amount of interaction with the environment in order to learn. In real-world environments, particularly when a risk factor is involved, an inept policy might be hazardous (Shalev-Shwartz et al., 2016). Thus, an appealing approach is to record a dataset of other agents in order to learn a safe initial policy which may later be improved via RL techniques (Taylor et al., 2011).

Learning from a dataset of experts has been extensively researched in the literature. These learning methods and algorithms are commonly referred to as Learning from Demonstrations (LfD) (Argall et al., 2009). In this paper, we consider a sibling, less explored problem of learning form a dataset of observations (LfO). We define LfO as a relaxation of LfD, where: (1) we do no assume a single policy generating the data and; (2) the policies are not assumed to be optimal, nor do they cover the entire state space. Practically, LfO is of interest since it is often easier to collect observations than expert demonstrations, for example, using crowdsourcing (Kurin et al., 2017). Technically, LfO is fundamentally different from LfD, where the typical task is to clone a single expert policy. The data under the LfO setting is expected to be more diverse than LfD data, which in general can be beneficial for learning. However, it also brings in the challenge of learning from multiple, possibly contradicting, policy trajectories.

In this work, we propose to solve the LfO problem with a three phases approach: (1) imitation (2) annotation and; (3) safe improvement. The imitation phase seeks to learn the average behavior in the dataset. In the annotation part we approximate the value function of the average behavior and in the final safe improvement step (section 5), we craft a novel algorithm that takes the learned average behavior and its approximated value function and yields an improved policy without generating new trajectories. The improvement step is designed to increase the policy performance with a high confidence in the presence of the value estimation errors that exists in a LfO setup.

Our three phases approach which we term Rerouted Behavior Improvement (RBI), provides a robust policy (without any interaction with the environment), that both eliminates the risks in random policy initialization and in addition, can boost the performance of a succeeding RL process. We demonstrate our algorithm both in a Taxi grid-world (Dietterich, 2000) as well as in the Atari domain (section 6). In the latter, we tackle the challenge of learning from non-experts human players

(Kurin et al., 2017). We show that our algorithm provides a robust policy, on par with deep RL policies, using only the demonstrations and without any additional interaction with the environment. As a baseline, we compare our approach to two state-of-the-art algorithms: (1) learning from demonstrations, DQfD (Hester et al., 2018) and; (2) robust policy improvement, PPO (Schulman et al., 2017).

## 2 RELATED WORK

Learning from demonstrations in the context of deep RL in the Atari environment have been studied in Cruz Jr et al. (2017), DQfD (Hester et al., 2018) and recently in Pohlen et al. (2018). However, all these methods focus on expert demonstrations and they benchmark their scores after additional trajectory collection with an iterative RL process. Therefore, essentially these methods can be categorized as a RL augmented with expert's supervised data. In contrast, we take a deeper look into the first part of best utilizing the observed data to provide the highest initial performance.

Previously, LfO has often been solved solely with the imitation step, e.g. in AlphaGo (Silver et al., 2016), where the system learned a policy which mimics the average behavior in a multiple policies dataset. While this provides sound empirical results, we found that one can do better by applying a safe improvement step to boost the policy performance. A greedy improvement method with respect to multiple policies has already been suggested in Barreto et al. (2017), yet we found that practically, estimating the value function of each different policy in the dataset is both computationally prohibitive and may also produce large errors since the generated data by each different policy is typically small. In section 4 we suggest a feasible alternative. Instead of learning the value of each different policy, estimate the value of the average behavior. While such estimation is not exact, we show both theoretically and experimentally that it provides a surrogate value function that may be used for policy improvement purposes.

There is also a significant research in the field of safe RL (García & Fernández, 2015), yet, here there may be multiple accepted definitions of this term ranging from worst-case criterion (Tamar et al., 2013) to baseline benchmark approach (Ghavamzadeh et al., 2016). We continue the line of research of safe RL investigated by Kakade & Langford (2002); Pirotta et al. (2013); Thomas et al. (2015) but we focus on a dataset composed of unknown policies. Finally, there are two recent well-known works in the context of non-decreasing policy improvement (also can be categorized as safe improvement) TRPO and PPO (Schulman et al., 2015; 2017). We compare our work to these algorithms and show two important advantages: First, our approach can be applied without an additional Neural Network (NN) policy optimization step and second we provide theoretical and experimental arguments why both approaches may be deemed unsafe when applied in the context of a LfO setup.

## 3 PROBLEM FORMULATION

We are dealing with a *Markov Decision Process* (MDP) (Puterman, 2014) where an agent interacts with an environment and tries to maximize a reward. A MDP is defined by the tuple $(\mathcal{S}, \mathcal{A}, \mathcal{P}, \mathcal{R})$, where $\mathcal{S}$ is a set of states and $\mathcal{A}$ is a set of actions. $\mathcal{P} : \mathcal{S} \times \mathcal{A} \to \mathcal{S}$ is the set of probabilities of switching from a state $s$ to $s'$ when executing action $a$, i.e. $P(s'|s, a)$ and $\mathcal{R}$ is a reward function $\mathcal{S} \times \mathcal{A} \to \mathbb{R}$ which defines the reward $r$ that the agent gets when applying action $a$ in state $s$. An agent acts according to a policy $\pi$, and its goal is to find a policy that maximizes the expected cumulative discounted reward, also known as the objective function $J(\pi) = \mathbb{E}\left[\sum_{k=0}^{\infty} \gamma^k r_k \big| s_0, \pi\right]$ where $\gamma < 1$ is a discount factor, $k$ is a time index and $s_0$ is an initial state.

We assume that all policies belong to the *Markovian randomized* set $\Pi^{MR}$ s.t. $\pi \in \Pi^{MR}$ is a probability distribution over $\mathcal{A}$ given a state $s$, i.e. $\pi(a|s)$.[1] For convenient and when appropriate, we may simply write $\pi_i$ to denote $\pi(a_i|s)$ (omitting the state's dependency). In the paper we will discuss two important distance measures between policies. The first is the Total Variation (TV) $\delta(\pi, \pi') = \frac{1}{2}\sum_{a_i}|\pi_i - \pi_i'|$ and the second is the KL divergence $D_{KL}(\pi||\pi') = -\sum_{a_i}\pi_i \log \frac{\pi_i'}{\pi_i}$.

---

[1] Note that humans' policies can generally be considered as part of the *history randomized* set $\Pi^{HR}$ where $\pi \in \Pi^{HR}$ is a probability function over $\mathcal{A}$ given the states and actions history. In the appendix we explain how we circumvented this hurdle in the Atari dataset.

These measures are often used to constrain the updates of a learned policy in an iterative policy improvement RL algorithm (Schulman et al., 2015; 2017).

For a given policy, the state's value is the expected cumulative reward starting at this state, $V^\pi(s) = \mathbb{E}\left[\sum_{k=0}^\infty \gamma^k r_k \big| s, \pi\right]$. Similarly, the $Q$-value function $Q^\pi(s, a)$ is the value of taking action $a$ in state $s$ and then immediately following with policy $\pi$. The advantage $A^\pi(s, a) = Q^\pi(s, a) - V^\pi(s)$ is the gain of taking action $a$ in state $s$ over the average value (note that $\sum_{a \in \mathcal{A}} \pi(a|s) A^\pi(s, a) = 0$). We denote by $P(s_0 \xrightarrow{k} s | \pi)$ the probability of switching from state $s_0$ to state $s$ in $k$ steps with a policy $\pi$.

We define the LfO problem as learning a policy $\pi$ solely by observing a finite set of trajectories of other behavior policies without interacting with the environment. Formally, we are given a dataset $\mathcal{D}$ of trajectories executed by $N$ different players each with a presumably different policy. Players are denoted by $p^i$, $i = 1, ..., N$, and their corresponding policies are $\beta^i$, with value and $Q$-value functions $V^i, Q^i$ respectively. $\mathcal{D}$ is indexed as $\{x_j\}_{j=0}^{|\mathcal{D}|}$, where the cardinality of the dataset is denoted by $|\mathcal{D}|$ and each record is the tuple $x_j = (s_j, a_j, r_j, t_j, i_j)$ s.t. $t_j$ is a termination signal and $i_j$ is the player's index. $\mathcal{D}$ may also be partitioned to $\mathcal{D} = \{\mathcal{D}_i | i = 1, .., N\}$, representing the different players' records.

The paper is accompanied with a running example, based on the Taxi grid-world domain (Dietterich, 2000), In the Taxi world, the driver's task is to pickup and drop a passenger in predefined locations with minimal number of steps (See figure 1). For this example, we synthetically generated policies of the form $\beta^i = \alpha^i(s)\pi^{rand} + (1 - \alpha^i(s))\pi^*$, where $\pi^*$ is the optimal policy and $\alpha^i(s)$ is a different mixing parameter for each different policy. Generally we divide the state space into two complementary and equal-size sets $\mathcal{S}_i^* \cup \bar{\mathcal{S}}_i^* = \mathcal{S}$, $|\mathcal{S}_i^*| = |\bar{\mathcal{S}}_i^*|$. Where $\alpha^i(s) = 0$ for $s \in \mathcal{S}_i^*$ and $\alpha^i(s) = 0.75$ for $s \in \bar{\mathcal{S}}_i^*$. In the next sections, we will use different selections of $\mathcal{S}_i^*$ to generate different types of datasets. For example, randomly pick half of the states to form $\bar{\mathcal{S}}_i^*$ is termed in the paper as a random selection (see also appendix).

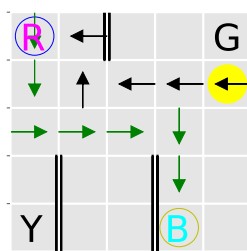

Figure 1: Taxi world

## 4 Average Behavior Policy and its Value Function

We begin with the first phase of the RBI approach which is learning a behavior policy from the dataset. Learning a behavior policy is a sensible approach both for generating a sound initial performance, as well as to avoid unknown states and actions. Yet, contrary to LfD where a single expert policy is observed, in multiple policies dataset the definition of a behavioral policy is ambiguous. To that end, we define the average behavior of the dataset as a natural generalization of the single policy imitation problem.

**Definition 4.1** (Average Behavior). The *average behavior* of a dataset $\mathcal{D}$ is

$$\beta(a|s) = \frac{\sum_{j \in \mathcal{D}} \mathbb{1}_{s,a}(j)}{\sum_{j \in \mathcal{D}} \mathbb{1}_s(j)} = \sum_{p^i} \frac{\sum_{j \in \mathcal{D}_i} \mathbb{1}_{s,a}(j)}{\sum_{j \in \mathcal{D}_i} \mathbb{1}_s(j)} \frac{\sum_{j \in \mathcal{D}_i} \mathbb{1}_s(j)}{\sum_{j \in \mathcal{D}} \mathbb{1}_s(j)}, \tag{1}$$

where $\mathbb{1}_{s,a}(j) = 1$ if $(s_j, a_j) = (s, a)$, and 0 otherwise. The first form in Eq. equation 4.1 is simply the fraction of each action taken in each state in $\mathcal{D}$, which for a single policy dataset is identical to behavioral cloning. Typically, when $\beta$ is expressed with a NN (as in section 6) we apply a standard classification learning process with a Cross Entropy loss. Otherwise, for a tabular representation (as in the Taxi example) we directly enumerate to calculate this expression. The second form in Eq. equation 4.1 is a weighted sum over all players in the dataset, which may also be expressed with conditional probability terminology as

$$\beta(a|s) = \sum_{p^i} \beta^i(a|s) P(p^i|s). \tag{2}$$

Here $P(p^i|s)$ is the probability of visiting an $i$th player's record given that a uniform sample $x \sim U(\mathcal{D})$ picked $s$. While other definitions of average behavior are possible, the ease of learning such

formulation with a NN makes it a natural candidate for RBI. Yet, for the second phase of RBI, one must evaluate its $Q$-value function, i.e. $Q^\beta$. To that end, we study two alternatives: (1) Temporal Difference (TD) learning and (2) Monte-Carlo (MC) approximation.

## 4.1 Learning $Q^\beta$ with TD Methods

A natural approach to evaluate $Q^\beta$, would be with off-policy TD learning (Sutton & Barto, 2017). However, directly learning $Q^\beta$ requires the knowledge of $\beta$ which can only be approximated in LfO. To skirt this requirement, we can first evaluate $V^\beta$ over the distribution of states in $\mathcal{D}$ with one-step TD. This is attained by minimizing the loss $E_{s\sim U(\mathcal{D})}\left[|V^\beta(s) - r(s,a) - \gamma V^\beta(s')|^2\right]$ where $s, a, s'$ are tuples of state-action and next state sampled from $\mathcal{D}$. Notice that since the distribution of actions in each state in the dataset equals to $\beta$, there is no requirement for Importance Sampling (IS) factor. With $V^\beta$ at hand, $Q^\beta$ can be learn by minimizing the loss

$$Q^\beta = \arg\min_{Q^\beta} E_{s,a,s'\sim U(\mathcal{D})}\left[|Q^\beta(s,a) - r(s,a) - \gamma V^\beta(s')|^2\right]. \tag{3}$$

While this TD method avoids IS calculation and doe not require any $\beta$ knowledge, it may still be problematic in LfO due to the bootstrapping operation. Bootstrapping is the practice of estimating the value of $s$ from the estimations of the next state $s'$. If our evaluation of $s'$ is incorrect, this error also propagates to $s$. Unlike, iterative RL this cannot be compensated by visiting again in $s'$ and updating the $s'$ value. In this sense, deep NN parametric forms are much more susceptible to bootstrapping since the evaluation error is larger comparing to a tabular representation (see also Sutton's Deadly Triad discussion in (Sutton & Barto, 2017)).

## 4.2 Approximating $Q^\beta$ with MC learning

As an alternative, to avoid bootstrapping we suggest to evaluate a surrogate function

$$Q^\mathcal{D}(s,a) = \sum_{p^i} Q^i(s,a)P\left(p^i|s,a\right) \tag{4}$$

which we term the *Q-value of the dataset*. $Q^\mathcal{D}$ may be interpreted as the weighted average over the players' $Q$-values, where the weights $P\left(p^i|s,a\right)$ are the probability of visiting an $i$th player's record given that a uniform sample $x \sim U(\mathcal{D})$, picked $s, a$. In the following two propositions, we show that such a function has two appealing characteristics. First, it may be evaluated with a $L_1$-norm loss and Monte-Carlo (MC) learning (Sutton & Barto, 2017) from the dataset trajectories, without off-policy corrections and without the burden of evaluating each $Q^i$ independently. Secondly, it is a $Q$-value function of a time dependent policy with a very similar structure to the average behavior. Taken together, they provide an efficient alternative to approximate the value function of $\beta$.

**Proposition 4.1** (Consistency of MC upper bound). *For an approximation $\hat{Q}^\mathcal{D}$ and a loss $\mathcal{L}_{L_1}(\hat{Q}^\mathcal{D}) = \frac{1}{|\mathcal{D}|}\sum_{j\in\mathcal{D}}|Q^\mathcal{D}(s_j,a_j) - \hat{Q}^\mathcal{D}(s_j,a_j)|$, an upper bound for the loss when $|\mathcal{D}|\to\infty$ is*

$$\mathcal{L}_{L_1}(\hat{Q}^\mathcal{D}) \leq \mathcal{L}_{MC}(\hat{Q}^\mathcal{D}) = \mathbb{E}_{x_j\sim U(\mathcal{D})}\left[\left|\hat{Q}^\mathcal{D}(s_j,a_j) - R_j\right|\right], \tag{5}$$

*where $R_j = \sum_{k\geq 0}\gamma^k r_{j+k}$ is the sampled Monte-Carlo return (Proof in the appendix).*

The attractive implication of Proposition 4.1 is that we can learn $Q^\mathcal{D}$ without any policy information and specifically without plugging Importance Sampling (IS) corrections such as $\frac{\beta}{\beta^i}$ which are known as a source of high variance (Munos et al., 2016). In addition, as suggested, this method does not use bootstrapping. The next proposition defines a policy with a $Q$-value equal to $Q^\mathcal{D}$.

**Proposition 4.2.** *Given a state-action pair $\tilde{s}, \tilde{a}$, then $Q^\mathcal{D}(\tilde{s},\tilde{a})$ is the Q-value of a time dependent policy $\beta_{\tilde{s},\tilde{a}}^\mathcal{D}(a|s,k)$, where $k$ is a time index and $\tilde{s}, \tilde{a}$ is a fixed initial state-action pair,*

$$\beta_{\tilde{s},\tilde{a}}^\mathcal{D}(a|s,k) = \sum_{p^i} \beta^i(a|s)P(p^i|\tilde{s},\tilde{a} \xrightarrow{k} s). \tag{6}$$

*Here the conditional probability is over a uniform sample $x \sim U(\mathcal{D}_{\tilde{s},\tilde{a},k})$ where $\mathcal{D}_{\tilde{s},\tilde{a},k}$ is a subset of $\mathcal{D}$ that contains all the entries in the dataset with distance $k$ from an entry with a state-action pair $\tilde{s}, \tilde{a}$ (Proof in the appendix).*

Proposition 4.2 indicates that when $P(p^i|\tilde{s}, \tilde{a} \xrightarrow{k} s)$ can be approximated as $P(p^i|s)$ at least for a finite horizon $k \propto \frac{1}{1-\gamma}$, then $Q^{\mathcal{D}} \simeq Q^{\beta}$. This happens when the distribution of players in states near $\tilde{s}, \tilde{a}$ equals to the distribution of players in those states in the entire dataset. Practically, while both policies are not equal, they have a relatively low TV distance and therefore their $Q$-values are close. To further increase the robustness of our method we add an interesting consideration: our improvement step will rely only on the *action ranking* of the $Q$-value in each state, i.e. the order of $\{Q^{\beta}(s, a_i)\}_{a_i \in \mathcal{A}}$ (see next section). This, as we show hereafter, significantly increases the effective similarity between $Q^{\mathcal{D}}$ and $Q^{\beta}$.

We demonstrates the action ranking similarity between $Q^{\beta}$ and $Q^{\mathcal{D}}$ in the Taxi grid-world example. To that end, we generated trajectories with $N$ players according to three selection types of $\mathcal{S}_i^*$: (1) row selection (2) column selection and; (3) random selection. Each selection type provides a different class of policies and therefore form a different dataset (see exact definitions in the appendix). In the first experiment (Figure 2a), we plot the average TV distance (for various initial states $\tilde{s}$), between $\beta$ and $\beta_{\tilde{s}, \tilde{a}}^{\mathcal{D}}$, as a function of the time-step for $N = 2$. Generally it is low, but it may be questionable whether relying on the true value of $Q^{\mathcal{D}}$ for the improvement step will provide adequate results. However, when we consider only the action ranking similarity (evaluated with the Pearson's rank correlation), we find even more favorable pattern.

First, in Figure 2b we plot the average rank correlation between $Q^{\beta}$ and $Q^{\mathcal{D}}$ (for $\gamma = 0.9$) as a function of the number of different policies used to generate the dataset. It is evident that the rank correlation is very high and stable for any number of policies. In the second experiment, we generated $N = 2$ (Figure 2c) and $N = 10$ (Figure 2d) policies and examined the impact of different discount factors. Also here, for the majority of practical scenarios we observe sufficiently high rank correlation. Only for a very large discount factor (close to 1) the rank correlation reduces. This happens since the long horizon accumulates more error from the difference (in high $k$ steps) in the TV distance. In conclusion, while Proposition 4.2 state bounds on the similarity between $Q^{\beta}$ and $Q^{\mathcal{D}}$, evaluating the Pearson's rank correlation confirms our statement that in practice the action ranking of $Q^{\mathcal{D}}$ is an acceptable surrogate for the action ranking of $Q^{\beta}$.

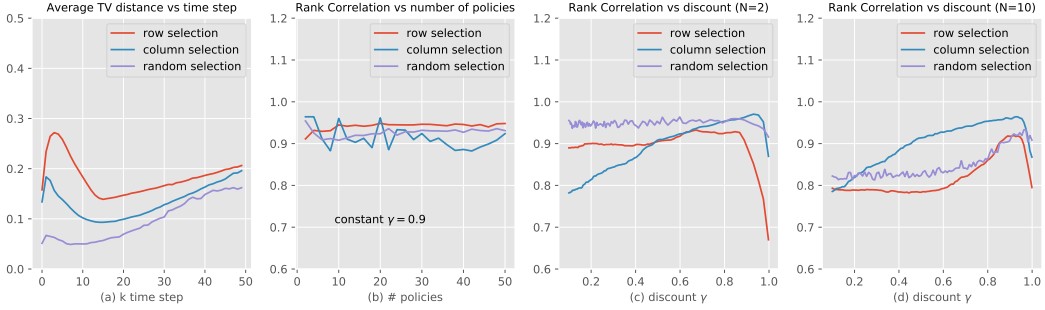

Figure 2: Taxi: comparison between $Q^{\beta}$ and $Q^{\mathcal{D}}$

## 5    SAFE POLICY IMPROVEMENT

Our next step, is to harness the approximated $Q$-function $Q^{\mathcal{D}} \simeq Q^{\beta}$ (which here will be termed $Q^{\beta}$), in order to improve the average behavior $\beta$, i.e. generate a policy $\pi$ such that $J(\pi) - J(\beta) \geq 0$. However, one must recall that $Q^{\beta}$ is learned from a fixed, sometimes even small dataset. Therefore, in order to guarantee improvement, we analyze the statistics of the value function's error. This leads to an interesting observation: the $Q$-value has a higher error for actions that were taken less frequently, thus, to avoid improvement penalty, we must restrict the ratio of the change in probability $c_i = \frac{\pi_i}{\beta_i}$. We will use this observation to craft our third phase of RBI, i.e. the safe improvement step, and show that other well-known monotonic improvement methods (such as PPO (Schulman et al., 2017) and TRPO (Schulman et al., 2015)) overlooked this consideration and therefore their improvement method may be unsafe for a LfO setup.

## 5.1 Soft Policy Improvement

Before analyzing the error's statistics, we begin by considering a subset of policies in $\Pi^{MR}$ which are proven to improve $\beta$ if our estimation of $Q^\beta$ is exact. Out of this subset we will later pick our improved policy. Recall that the most naive and also common improvement method is taking a greedy step, i.e. deterministically acting with the highest $Q$-value action in each state. This is known by the policy improvement theorem (Sutton & Barto, 2017), to improve the policy performance. The policy improvement theorem may be generalized to include a larger family of soft steps.

**Lemma 5.1** (Soft Policy Improvement). *Given a policy $\beta$, with value and advantage $V^\beta, A^\beta$, a policy $\pi \in \Pi^{MR}$ improves $\beta$, i.e. $V^\pi \geq V^\beta \ \forall s$, if it satisfies $\sum_a \pi(a|s)A^\beta(s,a) \geq 0 \ \forall s$ with at least one state with strict inequality. The term $\sum_a \pi(a|s)A^\beta(s,a)$ is called the improvement step.[2]*

Essentially, every policy that increases the probability of taking positive advantage actions over the probability of taking negative advantage actions achieves improvement. We will use the next Corollary to prove that our specific improvement step guarantees a positive improvement step.

**Corollary 5.1.1** (Rank-Based Policy Improvement). *Let $(A_i)_{i=1}^{|\mathcal{A}|}$ be an ordered list of the $\beta$ advantages in a state $s$, s.t. $A_{i+1} \geq A_i$, and let $c_i = \pi_i/\beta_i$. If for all states $(c_i)_{i=1}^{|\mathcal{A}|}$ is a monotonic non-decreasing sequence s.t. $c_{i+1} \geq c_i$, then $\pi$ improves $\beta$ (Proof in the appendix).*

## 5.2 Standard Error of the Value Estimation

To provide a statistical argument for the expected error of the $Q$-function, consider learning $Q^\beta$ with a tabular representation. The $Q$-function is the expected value of the random variable $z^\pi(s,a) = \sum_{k \geq 0} \gamma^k r_k | s, a, \pi$. Therefore, the Standard Error (SE) of an approximation $\hat{Q}^\beta(s,a)$ for the $Q$-value with $N$ MC trajectories is

$$\sigma_{\varepsilon(s,a)} = \frac{\sigma_{z(s,a)}}{\sqrt{N_s \beta(a|s)}}, \tag{7}$$

where $N_s = \sum_{j \in \mathcal{D}} \mathbb{1}_s(j)$ is the number of visitations in state $s$, s.t. $N = \beta(a|s)N_s$. Therefore, $\sigma_{\varepsilon(s,a)} \propto \frac{1}{\sqrt{\beta(a|s)}}$ and specifically for low frequency actions such estimation may suffer large SE.[3]

## 5.3 Policy Improvement in the Presence of Value Estimation Errors

We now turn to the crucial question of what happens when one applies an improvement step with respect to an inaccurate estimation of the $Q$-function, i.e. $\hat{Q}^\beta$.

**Lemma 5.2** (Improvement Penalty). *Let $\hat{Q}^\beta = \hat{V}^\beta + \hat{A}^\beta$ be an estimator of $Q^\beta$ with an error $\varepsilon(s,a) = (Q^\beta - \hat{Q}^\beta)(s,a)$ and let $\pi$ be a policy that satisfies lemma 5.1 with respect to $\hat{A}^\beta$. Then the following holds*

$$V^\pi(s) - V^\beta(s) \geq - \sum_{s' \in \mathcal{S}} \left( \sum_{k \geq 0} \gamma^k P(s \xrightarrow{k} s' | \pi) \right) \sum_{a \in \mathcal{A}} \varepsilon(s',a) \left( \beta(a|s') - \pi(a|s') \right), \tag{8}$$

*where the difference, denoted $\mathcal{E}(s)$, is called the improvement penalty (proof in the appendix).*

For simplicity we may write $\mathcal{E}(s) = - \sum_{s',a} \rho^\pi(s'|s)\varepsilon(s',a)(\beta(a|s') - \pi(a|s'))$, where $\rho^\pi(s'|s)$ is sometimes referred to as the undiscounted state distribution of policy $\pi$ given an initial state $s$. Since $\varepsilon(s',a)$ is a random variable, it is worth to consider the variance of $\mathcal{E}(s)$. Assuming that the errors $\varepsilon(s,a)$ are positively correlated (since neighboring state-action pairs share trajectories of rewards) and under the error model introduced above, it follows that

$$\sigma_{\mathcal{E}(s)}^2 \geq \sum_{s',a} (\rho^\pi(s'|s))^2 \sigma_{\varepsilon(s',a)}^2 (\beta(a|s') - \pi(a|s'))^2 = \sum_{s',a} \frac{(\rho^\pi(s'|s))^2 \sigma_{z(s',a)}^2}{N_{s'}} \frac{(\beta(a|s') - \pi(a|s'))^2}{\beta(a|s')}.$$

---

[2]We post a proof in the appendix for completeness, though it may have been proven elsewhere.

[3]Note that even for deterministic environments, a stochastic policy inevitably provides $\sigma_{z(s,a)} > 0$.

Hence, it is evident that the improvement penalty can be extremely large when the term $\frac{|\beta - \pi|^2}{\beta}$ is unregulated. Moreover, a single mistake along the trajectory, caused by an unregulated element, might wreck the performance of the entire policy. Therefore, it is essential to regulate each one of these elements to minimize the potential improvement penalty.

## 5.4 THE REROUTE CONSTRAINT

In order to regulate the ratio $\frac{|\beta - \pi|^2}{\beta}$, we suggest limiting the improvement step to a subset of $\Pi^{MR}$ based on the following constraint.

**Definition 5.1** (Reroute Constraint). Given a policy $\beta$, a policy $\pi$ is a $reroute(c_{\min}, c_{\max})$ of $\beta$, if $\pi(a|s) = c(s,a)\beta(a|s)$ where $c(s,a) \in [c_{\min}, c_{\max}]$. Further, note that reroute is a subset of the TV constraint with $\delta = \min(1 - c_{\min}, \max(\frac{c_{\max}-1}{2}, \frac{1-c_{\min}}{2}))$ *(proof in the appendix)*.

Now, it is evident that with the reroute constraint, each element in the sum of (8) is regulated and proportional to $\sqrt{\beta(a|s)}|1 - c(s,a)|$ where $c(s,a) \in [c_{\min}, c_{\max}]$. Analyzing other well-known trust regions such as the TV constraint $\delta \geq \frac{1}{2}\sum_a |\beta(a|s) - \pi(a|s)|$, the average KL-divergence constraint $\bar{D}_{KL}(\beta||\pi) = -\mathbb{E}_{s \sim \beta}[\sum_a \beta(a|s)\log\frac{\pi(a|s)}{\beta(a|s)}]$, used in the TRPO algorithm, and the PPO objective function (Schulman et al., 2017), surprisingly reveals that non of them properly controls the improvement penalty (see an example and an analysis of the PPO objective in the appendix, we also show in the appendix that the solution of the PPO objective is not unique).

## 5.5 MAXIMIZING THE IMPROVEMENT STEP UNDER THE REROUTE CONSTRAINT

We now consider the last part of our improvement approach, i.e. maximizing the objective function under the reroute constraint and whether such maximization provides a positive improvement step. It is well-known that maximizing the objective function without generating new trajectories of $\pi$ is a hard task since the distribution of states induced by the policy $\pi$ is unknown. Previous works have suggested to maximize a surrogate off-policy objective function $J^{OP}(\pi) = \mathbb{E}_{s \sim \beta}[\sum_a \pi(a|s)A^\beta(s,a)]$. These works have also suggested to solve the constrained maximization with a NN policy representation and a gradient ascent approach (Schulman et al., 2015). Here we suggest a refreshing alternative, instead of precalculating the policy $\pi$ that maximizes $J^{OP}$ one may ad hoc compute the policy that maximizes the improvement step $\sum_a \pi(a|s)A^\beta(s,a)$ (which is the argument of the $J^{OP}$ objective) for each different state. Such an approach maximizes also the $J^{OP}$ objective since the improvement step is independent between states. For the reroute constraint, this essentially sums up to solving the following simple linear program for each state

$$\text{Maximize: } (\boldsymbol{A}^\beta)^T\boldsymbol{\pi}$$
$$\text{Subject to: } c_{\min}\boldsymbol{\beta} \leq \boldsymbol{\pi} \leq c_{\max}\boldsymbol{\beta} \tag{9}$$
$$\text{And: } \sum \pi_i = 1.$$

Where $\boldsymbol{\pi}$, $\boldsymbol{\beta}$ and $\boldsymbol{A}^{\hat{\pi}}$ are vector representations of $(\pi(a_i|s))_{i=1}^{|\mathcal{A}|}$, $(\beta(a_i|s))_{i=1}^{|\mathcal{A}|}$ and $(A^\beta(s,a))_{i=1}^{|\mathcal{A}|}$ respectively. We term the algorithm that solves this maximization problem as Max-Reroute (see Algorithm 1). In the appendix we also provide an analogous algorithm that maximizes the improvement step under the TV constraint (termed Max-TV). We will use Max-TV as a baseline for the performance of the reroute constraint. With an ad hoc maximization approach, we avoid the hassle of additional learning task after the policy imitation step, and in addition, our solution guarantees maximization without the common caveats in NN learning such as converging to local minima or overfitting etc.

Further analyzing Max-Reroute and Max-TV quickly reveals that they both rely only on the action ranking at each state (as stated in the previous section). This is also in contrast with the aforementioned methods (TRPO and PPO) where by their definition as policy gradient methods (Sutton et al., 2000), they optimize the policy according to the magnitude of the advantage function. Finally, notice that both Max-Reroute and Max-TV satisfy the conditions of Corollary 5.1.1, therefore they always provide a positive improvement step and hence for a perfect approximation of the value function they are guaranteed to improve the performance.

---

**Algorithm 1:** Max-Reroute

---

**Data:** $s$, $\beta$, $A^\beta$, $(c_{\min}, c_{\max})$
**Result:** $\{\pi(a|s),\ a \in \mathcal{A}\}$
**begin**
  $\tilde{\mathcal{A}} \longleftarrow \mathcal{A}$
  $\Delta \longleftarrow 1 - c_{\min}$
  $\pi(a|s) \longleftarrow c_{\min}\beta(a|s)\ \forall a \in \mathcal{A}$
  **while** $\Delta > 0$ **do**
    $a = \arg\max_{a \in \tilde{\mathcal{A}}} A^\beta(s, a)$
    $\Delta_a = \min\{\Delta, (c_{\max} - c_{\min})\beta(a|s)\}$
    $\tilde{\mathcal{A}} \longleftarrow \tilde{\mathcal{A}}/a$
    $\Delta \longleftarrow \Delta - \Delta_a$
    $\pi(a|s) \longleftarrow \pi(a|s) + \Delta_a$

---

Let us now return to our Taxi example and examine different types of improvement steps with respect to a behavioral cloning baseline:: (1) a greedy step[4] (2) a TV constrained and; (3) a reroute constrained steps. The dataset is generated by two policies with a discount factor $\gamma = 0.9$. We consider two types of policies: row selection (3a) and random selection (3b). The behavior policy was calculated by enumerating actions in each state. We examine the two alternatives of evaluating $Q^\beta$: MC (without off-policy corrections) and TD learning (as described in section 4).

First, it is clear that taking a greedy step is a precarious approach. In these experiments, the generated greedy policy contained recurrent states (which is a repetitive series of states). This prevented the completion of the task. Comparing the TV step to the average behavior baseline, reveals that only in relatively large datasets (more than $10^3$ episodes in this example) TV step is better than behavioral cloning. This demonstrates the problematic approach of taking actions that were insufficiently evaluated. In real-world MDP with larger number of states, it is extremely difficult to sufficiently sample the entire state-space, hence, we project that TV should be poorer than behavioral cloning even for large datasets. In the next section, this premise is verified in the Atari domain.

Contrary to TV, reroute provided almost always better performance than the average behavior. This pattern repeats both with TD and MC learning. An important observation is that MC outperform TD for larger datasets. Since MC should converge to $Q^\mathcal{D}$ and TD to $Q^\beta$, we cannot link any performance degradation to the evaluation of $Q^\mathcal{D}$ instead of $Q^\beta$. On the other hand, we can still hypothesize that the slow improvement in TD (in large datasets) is due to bootstrapping. In the next section in the Atari example, we show that when we utilize a NN parametric form where bootstrapping errors are larger, this pattern intensifies.

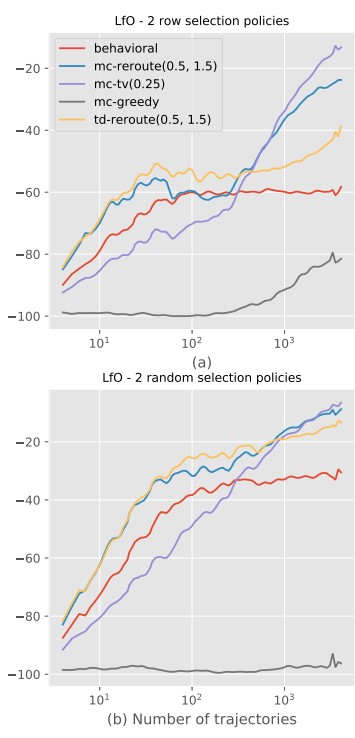

Figure 3: Taxi: Improvement steps comparison

## 6 LEARNING TO PLAY ATARI BY OBSERVING INEXPERT HUMAN PLAYERS

In the previous section, we analyzed the expected error which motivated the reroute constraint for a tabular representation. In this section, we experimentally show that the same ideas holds for Deep NN parametric form. We conducted an experiment with a crowdsourced data of 4 Atari games

---

[4]An unconstrained step, equivalent to reroute parameters $(c_{\min}, c_{\max}.) = (0, \infty)$

(Spaceinvaders, MsPacman, Qbert and Montezuma's Revenge) (Kurin et al., 2017). Each game had roughly 1000 recorded episodes. We employed two networks, one for policy cloning $\beta$ and one for $Q$-value estimation with architecture inspired by the Dueling DQN (Mnih et al., 2015; Wang et al., 2015) and a few modifications (see appendix). We evaluated two types of local maximization steps: Max-Reroute, with different parameters $(c_{\min}, c_{\max})$; and Max-TV. We implemented two baselines: (1) DQfD algorithm with hyperparameters as in Hester et al. (2018) and (2) a single PPO policy search step based on the learned behavior policy and the estimated advantage. The following discussion refers to the results presented in Figure 4.

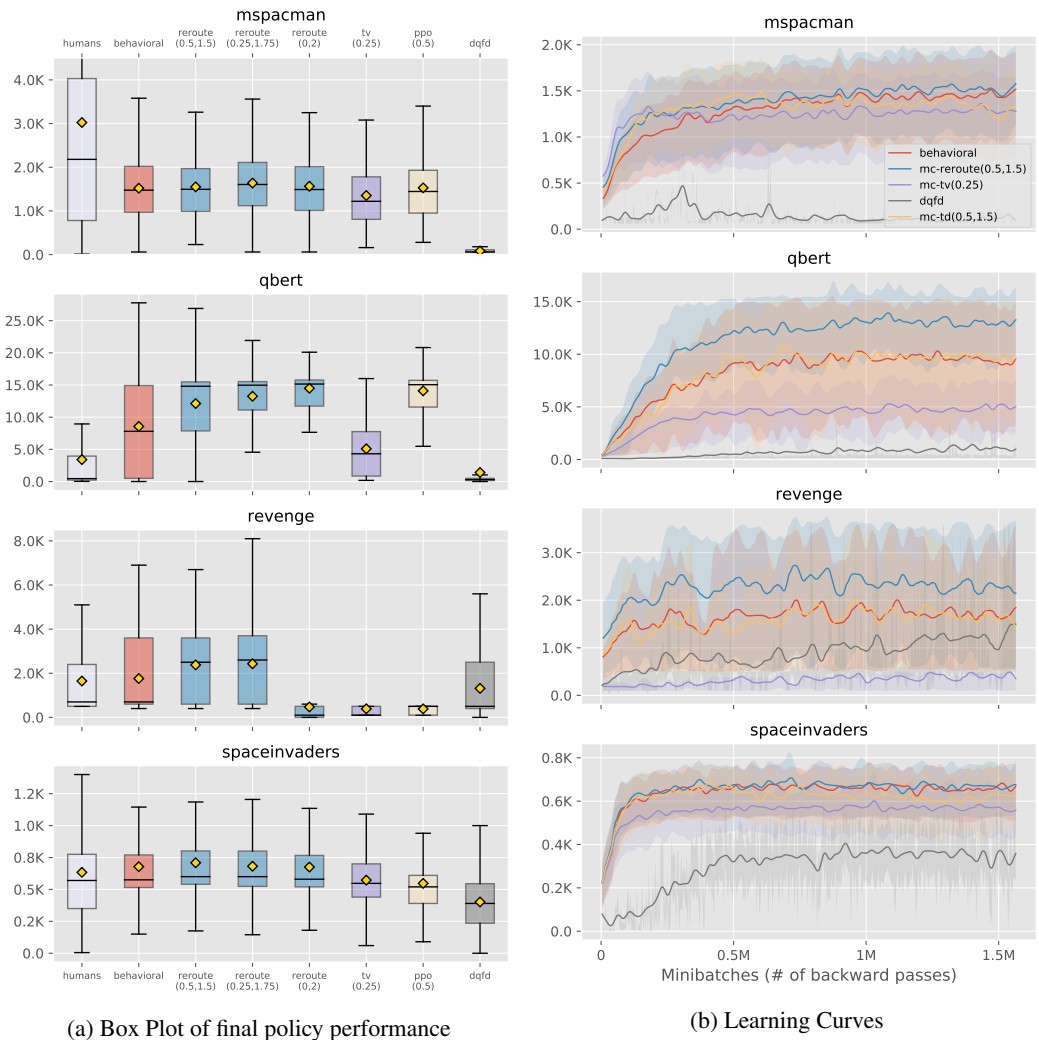

(a) Box Plot of final policy performance

(b) Learning Curves

Figure 4: Atari experiment results

**Average behavior performance:** When comparing the average behavior performance to the average human score, we get very similar performance in 2 out of 4 games. However, in Qbert, we obtained substantially better performance than the average human score. Generally, we expect that in games where longer trajectories lead to significantly more reward, the average policy obtains above average reward, since the effect of *good players* in the data has a heavier weight on the results. On the other hand, we assume that the lower score in MsPacman is due to the more complex and less clear frame in this game with respect to the other games. We take the average behavior performance as a baseline for comparing the safety level of the subsequent improvement steps.

**MC and TD estimators of the $Q$-value function:** We evaluated the performance of $reroute(0.5, 1.5)$ during the training process with two different estimators, both concurrently learned from the same batches. MC-$reroute(0.5, 1.5)$ learned the surrogate $Q^{\mathcal{D}}$ with full trajec-

tories and provided better score than the behavioral cloning baseline in 3 out of 4 games and tied in one. On the other hand TD-$reroute(0.5, 1.5)$ learned the true $Q^\beta$ with 1-step TD learning but did not show a significant improvement in any game. We hypothesize that larger evaluation errors due to bootstrapping may be the reason for the low performance of the TD estimator. The rest of the discussion focuses on the MC evaluation.

**Local maximization steps:** We chose to evaluate Max-TV with $\delta = 0.25$ since it encapsulates the $reroute(0.5, 1.5)$ region. It is clear that Max-TV is always dominated by Max-Reroute and it did not secure safe improvement in any of the games. A comparison of different reroute parameters reveals that there is an fundamental trade-off between safety and improvement. Smaller steps like $reroute(0.5, 1.5)$ support higher safety at the expense of improvement potential. In all games $reroute(0.5, 1.5)$ provided safe improvement, but in Qbert its results were inferior to $reroute(0, 2)$. On the other hand, $reroute(0, 2)$ reduced the Revenge score which indicates a too greedy step. Our results indicates that it is important to set $c_{\min} > 0$ since avoiding so may discard some important actions altogether due to errors in the value evaluation, resulting in a poor performance.

**Comparison with PPO:** For the PPO baseline, we executed a policy search with the learned advantage $A^\beta$ according to the PPO objective

$$J^{PPO}(\pi) = \mathbb{E}_{s \sim \beta} \left[ \sum_{a \in \mathcal{A}} \beta(a|s) \min \left( A^\beta(s, a) \frac{\pi(a|s)}{\beta(a|s)}, A^\beta(s, a) \operatorname{clip} \left( \frac{\pi(a|s)}{\beta(a|s)}, 1 - \varepsilon, 1 + \varepsilon \right) \right) \right].$$

We chose $\varepsilon = 0.5$, motivated by the similarity to the $reroute(0.5, 1.5)$ region (see appendix). Contrary to the PPO paper, we plugged our advantage estimator and did not use the Generalized Advantage Estimation (GAE). While $PPO(0.5)$ scored (see box plot in Figure 4a) better than $reroute(0.5, 1.5)$ in Qbert, in all other games it reduced the behavioral cloning score. The overall results indicate the similarity between $PPO(0.5)$ and $reroute(0, 2)$, probably since negative advantages actions tend to settle at zero-probability to avoid negative penalty. This emphasizes the importance of the $c_{\min}$ parameter of reroute which is missing from PPO.

**Comparison with DQfD:** DQfD scored below the average behavior in all games. The significantly low scores in MsPacman and Qbert raise the question whether DQfD and more generally, Off-Policy greedy RL can effectively learn from multiple non-exploratory *fixed policies*. The most conspicuous issue is the greedy policy improvement approach taken by DQfD: we have shown that an unconstrained greedy improvement step leads to a poor performance (recall the Taxi example). In addition, as discussed in section 4, Off-Policy RL with TD learning also suffers from the second RL ingredient, i.e. policy evaluation . As our results show, our proposed safe policy improvement scheme with MC learning mitigates these issues, leading to significantly better results

## 7    CONCLUSIONS

In this paper, we studied both theoretically and experimentally the problem of LfO. We analyzed factors that impede classical methods, such as TRPO/PPO and Off-Policy greedy RL, and proposed a novel alternative, Rerouted Behavior Improvement (RBI), that incorporates behavioral cloning and a safe policy improvement step. RBI is designed to learn from multiple agents and to mitigate value evaluation errors. It does not use importance sampling corrections or bootstrapping to estimate values, hence it is less sensitive to deep network function approximation errors. In addition, it does not require a policy search process. Our experimental results in the Atari domain demonstrate the strength of RBI compared to current state-of-the-art algorithms. We project that these attributes of RBI would also benefit an iterative RL process. Therefore, in the future, we plan to study RBI as an online RL policy improvement method.

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

# A  OPERATOR NOTATION

In part of the appendix, we use vector and operator notation, where $v^\pi$ and $q^\pi$ represent the value and $Q$-value functions as vectors in $\mathcal{V} \equiv \mathbb{R}^{|\mathcal{S}|}$ and $\mathcal{Q} \equiv \mathbb{R}^{|\mathcal{S} \times \mathcal{A}|}$. We denote the partial order $x \geq y \iff x(s,a) \geq y(s,a) \;\; \forall s, a \in \mathcal{S} \times \mathcal{A}$, and the norm $\|x\| = \max_{s,a} |x|$. Let us define two mappings between $\mathcal{V}$ and $\mathcal{Q}$:

1. $\mathcal{Q} \to \mathcal{V}$ mapping $\Pi^\pi$: $(\Pi^\pi q^\pi)(s) = \sum_{a \in \mathcal{A}} \pi(a|s) q^\pi(s,a) = v^\pi(s)$.
   This mapping is used to backup state-action-values to state value.

2. $\mathcal{V} \to \mathcal{Q}$ mapping $\tilde{\mathcal{P}}$: $\left(\tilde{\mathcal{P}} v^\pi\right)(s,a) = \sum_{s' \in \mathcal{S}} P(s'|s,a) v^\pi(s')$.
   This mapping is used to backup state-action-values based on future values. Note that this mapping is not dependent on a specific policy.

Let us further define two probability operators:

1. $\mathcal{V} \to \mathcal{V}$ state to state transition probability $P_v^\pi$:
   $(P_v^\pi v^\pi)(s) = \sum_{a \in \mathcal{A}} \pi(a|s) \sum_{s' \in \mathcal{S}} P(s'|s,a) v^\pi(s')$.

2. $\mathcal{Q} \to \mathcal{Q}$ state-action to state-action transition probability $P_q^\pi$:
   $\left(P_q^\pi q^\pi\right)(s,a) = \sum_{s' \in \mathcal{S}} P(s'|s,a) \sum_{a' \in \mathcal{A}} \pi(a'|s') q^\pi(s',a')$.

Note that the probability operators are bounded linear transformations with spectral radios $\sigma(P_v^\pi) \leq 1$ and $\sigma(P_q^\pi) \leq 1$ (Puterman, 2014). Prominent operators in the MDP theory are the recursive value operator $T_v^\pi v = \Pi^\pi r + \gamma P_v^\pi v$ and recursive $Q$-value operator $T_q^\pi q = r + \gamma P_q^\pi q$. Both have a unique solution for the singular value equations $v = T_v^\pi v$ and $q = T_q^\pi q$. These solutions are $v^\pi$ and $q^\pi$ respectively (Sutton & Barto, 2017; Puterman, 2014). In addition, in the proofs we will use the following easy to prove properties:

1. $P_q^\pi = \tilde{\mathcal{P}} \Pi^\pi$

2. $P_v^\pi = \Pi^\pi \tilde{\mathcal{P}}$

3. $x \geq y, \;\; x, y \in \mathcal{V}, \Rightarrow \tilde{\mathcal{P}} x \geq \tilde{\mathcal{P}} y$

4. The probability of transforming from $s$ to $s'$ in $k$ steps is $P(s \xrightarrow{k} s' | \pi) = (P_v^\pi)^k(s,s')$.

5. $[(P_v^\pi)^k x](s) = \mathbb{E}_{s_k \sim \pi|s} [x(s_k)]$

# B  AVERAGE BEHAVIOR PROOFS

## B.1  UPPER BOUND FOR MONTE-CARLO EVALUATION OF $Q^\mathcal{D}$ WITH $L_1$ LOSS

*Proof.* We denote by $\mathcal{L}_{L_1}(\hat{Q}^\mathcal{D})$ the true $L_1$ least absolute error loss of the value function estimator, and we show that $\mathcal{L}_{MC}(\hat{Q}^\mathcal{D}) \geq \mathcal{L}_{L_1}(\hat{Q}^\mathcal{D})$ for $|\mathcal{D}| \to \infty$ such that minimization of $\mathcal{L}_{MC}$ leads to a bounded error of $\mathcal{L}_{L_1}$. Remember that the dataset $\mathcal{D}$ is a concatenation of the sets $\mathcal{D}_i = \{(s_1, a_1), (s_2, a_2), ...\}$, $i = 1, .., N$.

$$\mathcal{L}_{L_1}(\hat{Q}^\mathcal{D}) = \frac{1}{|\mathcal{D}|} \sum_{j \in \mathcal{D}} \left| Q^\mathcal{D}(s_j, a_j) - \hat{Q}^\mathcal{D}(s_j, a_j) \right| =$$

$$\frac{1}{|\mathcal{D}|} \sum_{j \in \mathcal{D} \cap \Omega^<} \left[ \sum_{p^i} Q^i(s_j, a_j) P\left(p^i | s_j, a_j\right) - \hat{Q}^\mathcal{D}(s_j, a_j) \right] +$$

$$+ \frac{1}{|\mathcal{D}|} \sum_{j \in \mathcal{D} \cap \Omega^>} \left[ \hat{Q}^\mathcal{D}(s_j, a_j) - \sum_{p^i} Q^i(s_j, a_j) P\left(p^i | s_j, a_j\right) \right]$$

where $\Omega^<$ and $\Omega^>$ denote the complementary subspaces of $\mathcal{S} \times \mathcal{A}$ where $\hat{Q}^{\mathcal{D}}$ underestimates and overestimates the target (respectively).

$$\frac{1}{|\mathcal{D}|} \sum_{j \in \mathcal{D} \cap \Omega^<} \left[ \sum_{p^i} Q^i(s_j, a_j) P\left(p^i | s_j, a_j\right) - \hat{Q}^{\mathcal{D}}(s_j, a_j) \right] =$$

$$\frac{1}{|\mathcal{D}|} \sum_{j \in \mathcal{D} \cap \Omega^<} \sum_{p^i} P\left(p^i | s_j, a_j\right) \left[ Q^i(s_j, a_j) - \hat{Q}^{\mathcal{D}}(s_j, a_j) \right] \leq$$

$$\frac{1}{|\mathcal{D}|} \sum_{j \in \mathcal{D} \cap \Omega^<} \sum_{p^i} P\left(p^i | s_j, a_j\right) \left| Q^i(s_j, a_j) - \hat{Q}^{\mathcal{D}}(s_j, a_j) \right|.$$

Joining the equivalent term for the $\Omega^>$ region, we obtain the inequality

$$\mathcal{L}_{L_1}(\hat{Q}^{\mathcal{D}}) \leq \frac{1}{|\mathcal{D}|} \sum_{j \in \mathcal{D}} \sum_{p^i} P\left(p^i | s_j, a_j\right) \left| Q^i(s_j, a_j) - \hat{Q}^{\mathcal{D}}(s_j, a_j) \right| =$$

$$\frac{1}{|\mathcal{D}|} \sum_{unique(s,a) \in \mathcal{D}} \left( \sum_{j \in \mathcal{D}} \mathbb{1}_{s,a}(j) \right) \sum_{p^i} \frac{\sum_{j \in \mathcal{D}_i} \mathbb{1}_{s,a}(j)}{\sum_{j \in \mathcal{D}} \mathbb{1}_{s,a}(j)} \left| Q^i(s_j, a_j) - \hat{Q}^{\mathcal{D}}(s_j, a_j) \right| =$$

$$\frac{1}{|\mathcal{D}|} \sum_{unique(s,a) \in \mathcal{D}} \sum_{p^i} \left( \sum_{j \in \mathcal{D}_i} \mathbb{1}_{s,a}(j) \right) \left| Q^i(s_j, a_j) - \hat{Q}^{\mathcal{D}}(s_j, a_j) \right| =$$

$$\frac{1}{|\mathcal{D}|} \sum_{j \in \mathcal{D}} \left| Q^{i_j}(s_j, a_j) - \hat{Q}^{\mathcal{D}}(s_j, a_j) \right| = \frac{1}{|\mathcal{D}|} \sum_{j \in \mathcal{D}} \left| \mathbb{E}\left[ R_j | s_j, a_j, p^{i_j} \right] - \hat{Q}^{\mathcal{D}}(s_j, a_j) \right|$$

Plugging in the Jensen inequality for all $s_j, a_j, p^{i_j}$

$$\left| \mathbb{E}\left[ R_j | s_j, a_j, p^{i_j} \right] - \hat{Q}^{\mathcal{D}}(s_j, a_j) \right| \leq \mathbb{E}\left[ |R_j - \hat{Q}^{\mathcal{D}}(s_j, a_j)| \Big| s_j, a_j, p^{i_j} \right]$$

we obtain

$$\mathcal{L}_{L_1}(\hat{Q}^{\mathcal{D}}) \leq \frac{1}{|\mathcal{D}|} \sum_{j \in \mathcal{D}} \mathbb{E}\left[ |R_j - \hat{Q}^{\mathcal{D}}(s_j, a_j)| \Big| s_j, a_j, p^{i_j} \right].$$

In the limit when $|\mathcal{D}| \to \infty$ the sample average converges to the expectation, i.e. $\frac{1}{|\mathcal{D}|} \sum_{j \in \mathcal{D}} [\cdot] \to \mathbb{E}[\cdot]$, therefore

$$\mathcal{L}_{L_1}(\hat{Q}^{\mathcal{D}}) \leq \frac{1}{|\mathcal{D}|} \sum_{j \in \mathcal{D}} \mathbb{E}\left[ |R_j - \hat{Q}^{\mathcal{D}}(s_j, a_j)| \Big| s_j, a_j, p^{i_j} \right]$$

$$= \mathbb{E}\left[ \mathbb{E}\left[ |R_j - \hat{Q}^{\mathcal{D}}(s_j, a_j)| \Big| s_j, a_j, p^{i_j} \right] \right]$$

$$= \mathbb{E}\left[ |R_j - \hat{Q}^{\mathcal{D}}(s_j, a_j)| \right]$$

$$= \frac{1}{|\mathcal{D}|} \sum_{j \in \mathcal{D}} |R_j - \hat{Q}^{\mathcal{D}}(s_j, a_j)| = \mathcal{L}_{MC}(\hat{Q}^{\mathcal{D}}).$$

$\square$

## B.2 $Q^{\mathcal{D}}$ IS A VALUE OF A TIME DEPENDENT POLICY $\beta_{\tilde{s},\tilde{a}}^{\mathcal{D}}$

Given a state-action pair $\tilde{s}, \tilde{a}$, then $Q^{\mathcal{D}}(\tilde{s}, \tilde{a})$ is the $Q$-value of a time dependent policy $\beta_{\tilde{s},\tilde{a}}^{\mathcal{D}}(a|s, k)$, where $k$ is a time index and $\tilde{s}, \tilde{a}$ is a fixed initial state-action pair,

$$\beta_{\tilde{s},\tilde{a}}^{\mathcal{D}}(a|s, k) = \sum_{p^i} \beta^i(a|s) P(p^i | \tilde{s}, \tilde{a} \xrightarrow{k} s). \tag{10}$$

Here the conditional probability is over a uniform sample $x \sim U(\mathcal{D}_{\tilde{s},\tilde{a},k})$ where $\mathcal{D}_{\tilde{s},\tilde{a},k}$ is a subset of $\mathcal{D}$ that contains all the entries in the dataset with distance $k$ from an entry with a state-action pair $\tilde{s}, \tilde{a}$.

*Proof.* For the proof, let us construct an extended MDP, with an additional initial state, termed $s_0$ with $N$ different actions. Following an action selection in state $s_0$, an action $\tilde{a}$ is executed in state $\tilde{s}$ (in the original MDP) and then the original MDP continues as usual.

Let us define a History Randomized (HR) policy $\beta^h \in \Pi^{HR}$ according to the following rule:

1. In state $s_0$, select an action $i$ with probability $P(p^i|\tilde{s}, \tilde{a})$.

2. In all consecutive states act with the Markovian policy $\beta^i$.

Notice that it is clearly a HR policy since the second step is determined by the history (that is the first step). It is straightforward to calculate the value of the $s_0$ state with the law of total probability s.t. $V^{\beta^h}(s_0) = \sum_{\pi} Q^i(\tilde{s}, \tilde{a})P(p^i|\tilde{s}, \tilde{a}) = Q^{\mathcal{D}}(\tilde{s}, \tilde{a})$.

Puterman (2014) proved (Theorems 5.5.1-3) that for every HR policy, there exists a time dependent Markovian policy with the same value, let it be denoted as $\beta^{\mathcal{D}}_{\tilde{s},\tilde{a}}$. The Markovian policy satisfies

$$\beta^{\mathcal{D}}_{\tilde{s},\tilde{a}}(a|s, k) = P(a|s, k, \beta^h),$$

that is the probability of executing action $a$ in state $s$ and time step $k$ when following $\beta^h$. Again, with the law of total probability this may be written as

$$\beta^{\mathcal{D}}_{\tilde{s},\tilde{a}}(a|s, k) = \sum_i \beta^i(a|s)P(a(s_0) = i|s_0 \xrightarrow{k} s),$$

where $a(s_0)$ is the chosen action in state $s_0$. Notice that $\tilde{s}, \tilde{a}$ deterministically follow state $s_0$ so by applying Bayes' theorem, we get that

$$P(a(s_0) = i|s_0 \xrightarrow{k} s) = \frac{P(a(s_0) = i)P(\tilde{s}, \tilde{a} \xrightarrow{k} s|a(s_0) = i)}{P(\tilde{s}, \tilde{a} \xrightarrow{k} s)} = \frac{P(p^i|\tilde{s}, \tilde{a})P(\tilde{s}, \tilde{a} \xrightarrow{k} s|\beta^i)}{\sum_{p^j} P(p^j|\tilde{s}, \tilde{a})P(\tilde{s}, \tilde{a} \xrightarrow{k} s|\beta^j)}.$$

Let us define by $\mathbb{1}_{\tilde{s},\tilde{a},k,s}$ an indicator function which equals 1 for the entries in the dataset which are in state $s$ in a distance $k$ from an entry with state-action pair $\tilde{s}, \tilde{a}$. We may represent the above expression with the indicator summation notation

$$\frac{P(p^i|\tilde{s}, \tilde{a})P(\tilde{s}, \tilde{a} \xrightarrow{k} s|\beta^i)}{\sum_{p^j} P(p^j|\tilde{s}, \tilde{a})P(\tilde{s}, \tilde{a} \xrightarrow{k} s|\beta^j)} = \frac{\frac{\sum_{j\in\mathcal{D}^i} \mathbb{1}_{\tilde{s},\tilde{a}}}{\sum_{j\in\mathcal{D}} \mathbb{1}_{\tilde{s},\tilde{a}}} \frac{\sum_{j\in\mathcal{D}^i} \mathbb{1}_{\tilde{s},\tilde{a},k,s}}{\sum_{j\in\mathcal{D}^i} \mathbb{1}_{\tilde{s},\tilde{a}}}}{\sum_{p^{i'}} \left( \frac{\sum_{j\in\mathcal{D}^{i'}} \mathbb{1}_{\tilde{s},\tilde{a}}}{\sum_{j\in\mathcal{D}} \mathbb{1}_{\tilde{s},\tilde{a}}} \frac{\sum_{j\in\mathcal{D}^{i'}} \mathbb{1}_{\tilde{s},\tilde{a},k,s}}{\sum_{j\in\mathcal{D}^{i'}} \mathbb{1}_{\tilde{s},\tilde{a}}} \right)} = \frac{\sum_{j\in\mathcal{D}^i} \mathbb{1}_{\tilde{s},\tilde{a},k,s}}{\sum_{j\in\mathcal{D}} \mathbb{1}_{\tilde{s},\tilde{a},k,s}}.$$

Finally, the last expression can be transformed back to conditional probability notation of the form

$$\frac{\sum_{j\in\mathcal{D}^i} \mathbb{1}_{\tilde{s},\tilde{a},k,s}}{\sum_{j\in\mathcal{D}} \mathbb{1}_{\tilde{s},\tilde{a},k,s}} = P(p^i|\tilde{s}, \tilde{a} \xrightarrow{k} s)$$

$\square$

## C  SAFE POLICY IMPROVEMENT

### C.1  REROUTE IS A SUBSET OF TV

The set of reroute policies with $[c_{\min}, c_{\max}]$ is a subset of the set of $\delta$-TV policies, where $\delta = \min(1 - c_{\min}, \max(\frac{c_{\max}-1}{2}, \frac{1-c_{\min}}{2}))$.

*Proof.* For a reroute policy, the TV distance is $\frac{1}{2} \sum_a \beta(a_i|s)|1 - c_i| \leq \sum_a \beta(a_i|s)\delta = \delta$. For the second upper bound $1 - c_{\min}$, notice that a rerouted policy $\{\pi_i\}$ may be written as $\pi_i = c_{\min}\beta_i + \delta_i$, where $\delta_i \geq 0$. Therefore, the TV distance is

$$TV = \frac{1}{2}\sum_i |\pi_i - \beta_i| = \frac{1}{2}\sum_i |c_{\min}\beta_i + \delta_i - \beta_i| \leq \frac{1}{2}\sum_i ((1 - c_{\min})\beta_i + \delta_i),$$

but $1 = \sum_i \pi_i = \sum_i (c_{\min}\beta_i + \delta_i) = c_{\min} + \sum_i \delta_i$. Thus, $\sum_i \delta_i = 1 - c_{\min}$ and we may write

$$TV \leq \frac{1}{2}\left((1 - c_{\min}) + (1 - c_{\min})\right) = 1 - c_{\min}.$$

To show that the TV region is not in reroute, take any policy with a zero probability action and another action with a probability of $\beta(a_i|s) \leq \delta$. The policy which switches probabilities between these two actions is in $TV(\delta)$ but it is not in reroute for any finite $c_{\max}$. □

## C.2 SOFT POLICY IMPROVEMENT

The soft policy improvement rule states that every policy $\pi$ that satisfies

$$\sum_a \pi(a|s)A^\beta(s,a) \geq 0 \ \forall s \in \tilde{\mathcal{S}}$$

improves the policy $\beta$ s.t. $V^\pi \geq V^\beta \ \forall s$. To avoid stagnation we demand that the inequality be strict for at least a single state. In operator notation, the last equation can be written as

$$\Pi^\pi a^\beta \geq 0$$

*Proof.* Plugging in $a^\beta = q^\beta - v^\beta$ and using $v^\beta = \Pi^\pi v^\beta$, we get that

$$v^\beta \leq \Pi^\pi q^\beta = \Pi^\pi (r + \gamma \tilde{\mathcal{P}} v^\beta) = r^\pi + \gamma P_v^\pi v^\beta. \tag{11}$$

Then, by applying (11) recursively we get

$$v^\beta \leq r^\pi + \gamma P_v^\pi v^\beta = r^\pi + \gamma P_v^\pi (r^\pi + \gamma P_v^\pi v^\beta) \leq r^\pi + \gamma P_v^\pi r^\pi + \gamma^2 (P_v^\pi)^2 v^\beta \leq r^\pi +$$
$$\gamma P_v^\pi r^\pi + \gamma^2 (P_v^\pi)^2 r^\pi + \gamma^3 (P_v^\pi)^3 v^\beta \leq ... \leq \sum_{k \geq 0} \gamma^k (P_v^\pi)^k r^\pi = (\mathcal{I} - \gamma P_v^\pi)^{-1} r^\pi = v^\pi.$$

□

(Schulman et al., 2015) have shown that

$$J(\pi) - J(\beta) = \mathbb{E}_{T \sim \pi}\left[\sum_{k \geq 0} \gamma^k A^\beta(s_k, a_k)\right].$$

This equation can also be written as

$$J(\pi) - J(\beta) = \sum_{k \geq 0} \gamma^k \mathbb{E}_{s_k \sim \pi|s_0}\left[\sum_{a \in \mathcal{A}} \pi(a|s_k)A^\beta(s_k, a_k)\right] =$$
$$= \sum_{k \geq 0} \gamma^k \sum_{s \in \mathcal{S}} P(s_0 \xrightarrow{k} s|\pi) \sum_{a \in \mathcal{A}} \pi(a|s)A^\beta(s, a) =$$
$$\sum_{s \in \mathcal{S}}\left(\sum_{k \geq 0} \gamma^k P(s_0 \xrightarrow{k} s|\pi)\right) \sum_{a \in \mathcal{A}} \pi(a|s)A^\beta(s_k, a).$$

Recall the definition of the discounted distribution of states

$$\rho^\pi(s) = \sum_{k \geq 0} \gamma^k P(s_0 \xrightarrow{k} s|\pi),$$

we conclude that

$$J(\pi) - J(\beta) = \sum_{s \in \mathcal{S}} \rho^\pi(s) \sum_{a \in \mathcal{A}} \pi(a|s)A^\beta(s, a)$$

## C.3 RANK-BASED POLICY IMPROVEMENT

*Proof.* Denote by $i$ the index of the advantage ordered list $\{A_i\}$. Since $\sum_{a_i} \beta_i A_i^\beta = 0$, we can write it as an equation of the positive and negative advantage components

$$\sum_{a=i_p}^{N} \beta_i A_i^\beta = \sum_{a=0}^{i_n} \beta_i(-A_i^\beta),$$

where $i_p$ is the minimal index of positive advantages and $i_n$ is the maximal index of negative advantages. Since all probability ratios are non-negative, it is sufficient to show that

$$\sum_{a=i_p}^{N} c_i \beta_i A_i^\beta \geq \sum_{a=0}^{i_n} c_i \beta_i(-A_i^\beta).$$

But clearly

$$\sum_{a=i_p}^{N} c_i \beta_i A_i^\beta \geq c_{i_p} \sum_{a=i_p}^{N} \beta_i A_i^\beta \geq c_{i_n} \sum_{a=0}^{i_n} \beta_i(-A_i^\beta) \geq \sum_{a=0}^{i_n} c_i \beta_i(-A_i^\beta).$$

$\square$

## C.4 POLICY IMPROVEMENT PENALTY

Let $\hat{Q}^\beta$ be an approximation of $Q^\beta$ with an error $\varepsilon(s,a) = (Q^\beta - \hat{Q}^\beta)(s,a)$ and let $\pi$ be a policy that satisfies the soft policy improvement theorem with respect to $\hat{Q}^\beta$. Then the following holds:

$$V^\pi(s) - V^\beta(s) \geq -\sum_{s' \in \mathcal{S}} \left( \sum_{k \geq 0} \gamma^k P(s \xrightarrow{k} s' | \pi) \right) \sum_{a \in \mathcal{A}} \varepsilon(s',a) \left( \beta(a|s') - \pi(a|s') \right). \qquad (12)$$

The proof resembles the Generalized Policy improvement theorem (Barreto et al., 2017).

*Proof.* We will use the equivalent operator notation. Define the vector $\varepsilon = q^\beta - \hat{q}^\beta$

$$\begin{aligned}
T_q^\pi \hat{q}^\beta = r + \gamma \tilde{\mathcal{P}} \Pi^\pi \hat{q}^\beta &\geq r + \gamma \tilde{\mathcal{P}} \Pi^\beta \hat{q}^\beta \\
&= r + \gamma \tilde{\mathcal{P}} \Pi^\beta q^\beta - \gamma \tilde{\mathcal{P}} \Pi^\beta \varepsilon \\
&= T_q^\beta q^\beta - \gamma P_q^\beta \varepsilon \\
&= q^\beta - \gamma P_q^\beta \varepsilon \\
&= \hat{q}^\beta + (1 - \gamma P_q^\beta) \varepsilon
\end{aligned}$$

.

Note that the inequality is valid, since $\Pi^\pi \hat{q}^\beta \geq \Pi^\beta \hat{q}^\beta$ (by the theorem assumptions), and if $v \geq u$ then $\tilde{\mathcal{P}} v \geq \tilde{\mathcal{P}} u$. Set $y = (1 - \gamma P_q^\beta)\varepsilon$ and notice that $T_q^\pi(\hat{q}^\beta + y) = T_q^\pi \hat{q}^\beta + \gamma P_q^\pi y$. By induction, we show that

$$(T_q^\pi)^n \hat{q}^\beta \geq \hat{q}^\beta + \sum_{k=0}^{n} \gamma^k (P_q^\pi)^k y.$$

We showed it for $n = 1$, assume it holds for $n$, then for $n+1$ we obtain

$$(T_q^\pi)^{n+1} \hat{q}^\beta = T_q^\pi (T_q^\pi)^n \hat{q}^\beta \geq T_q^\pi \left( \hat{q}^\beta + \sum_{k=0}^{n} \gamma^k (P_q^\pi)^k y \right) =$$

$$= T_q^\pi \hat{q}^\beta + \sum_{k=0}^{n} \gamma^{k+1} (P_q^\pi)^{k+1} y \geq \hat{q}^\beta + y + \sum_{k=0}^{n} \gamma^{k+1} (P_q^\pi)^{k+1} = \hat{q}^\beta + \sum_{k=0}^{n+1} \gamma^k (P_q^\pi)^k y.$$

Using the contraction properties of $T_q^\pi$, s.t. $\lim_{k\to\infty}(T_q^\pi)^k x = q^\pi$, $\forall x \in \mathcal{Q}$ and plugging back $(1 - \gamma P_q^\beta)\varepsilon = y$, we obtain

$$q^\pi = \lim_{n\to\infty}(T_q^\pi)^n(\hat{q}^\beta) \geq \hat{q}^\beta + \sum_{k\geq 0}\gamma^k(P_q^\pi)^k(1-\gamma P_q^\beta)\varepsilon$$

$$= \hat{q}^\beta + \varepsilon + \sum_{k\geq 0}\gamma^{k+1}(P_q^\pi)^k(P_q^\pi - P_q^\beta)\varepsilon.$$

Applying $\Pi^\pi$ we transform back into $\mathcal{V}$ space

$$v^\pi \geq \Pi^\pi\hat{q}^\beta + \Pi^\pi\varepsilon + \sum_{k\geq 0}\gamma^{k+1}\Pi^\pi(P_q^\pi)^k(P_q^\pi - P_q^\beta)\varepsilon$$

$$\geq \Pi^\beta\hat{q}^\beta + \Pi^\pi\varepsilon + \sum_{k\geq 0}\gamma^{k+1}\Pi^\pi(P_q^\pi)^k(P_q^\pi - P_q^\beta)\varepsilon$$

$$= v^\beta + (\Pi^\pi - \Pi^\beta)\varepsilon + \sum_{k\geq 0}\gamma^{k+1}\Pi^\pi(P_q^\pi)^k(P_q^\pi - P_q^\beta)\varepsilon$$

.

Notice that $\Pi^\pi(P_q^\pi)^k P_q^\pi = \Pi^\pi(\tilde{\mathcal{P}}\Pi^\pi)^k\tilde{\mathcal{P}}\Pi^\pi = (\Pi^\pi\tilde{\mathcal{P}})^{k+1}\Pi^\pi = (P_v^\pi)^{k+1}\Pi^\pi$, and in the same manner, $\Pi^\pi(P_q^\pi)^k P_q^\beta = (P_v^\pi)^{k+1}\Pi^\beta$. Therefore, we can write

$$v^\pi \geq v^\beta - \sum_{k\geq 0}\gamma^k(P_v^\pi)^k(\Pi^\beta - \Pi^\pi)\varepsilon,$$

which may also be written as (12). □

## C.5 Unbounded Probability Ratios in the TV and KL constraints

To verify that TV and KL do not regulate the probability ratios, let's consider a tiny example of maximizing the objective function $J^{\tilde{P}PO}$ for a single state MDP with two actions $\{a_0, a_1\}$. Assume a behavior policy $\beta = \{1, 0\}$ and an estimated advantage $A^\beta = \{0, 1\}$. We search for a policy $\pi = \{1 - \alpha, \alpha\}$ that Maximize improvement step under the TV or KL constraints.

For a $\delta$-TV constraint, $\frac{1}{2}(1 - (1 - \alpha) + \alpha) = \alpha \leq \delta$. The improvement step in this case is $\sum_{a_i} A_i^\beta \pi_i = \alpha$. Hence the solution is $\alpha = \delta$ and the probability ratio $\pi(a_1)/\beta(a_1)$ is unconstrained (and undefined). Similarly for a $\delta$-KL constraint we get $-\log\alpha \leq \delta$ and the improvement step is identical. Hence $\alpha = e^{-\delta}$ and again, no constraint is posed on the probability ratio.

## C.6 Max-TV

---
**Algorithm 2:** Max-TV
---
**Data:** $s$, $\beta$, $A^\mathcal{D}$, $\delta$
**Result:** $\{\pi(a|s),\ a \in \mathcal{A}\}$
**begin**
    $\pi(a|s) \longleftarrow \beta(a|s), \forall a \in \mathcal{A}$
    $a = \arg\max_{a\in\tilde{\mathcal{A}}} A^\mathcal{D}(s, a)$
    $\Delta = \min\{\delta, 1 - \beta(a|s)\}$
    $\pi(a|s) \longleftarrow \pi(a|s) + \Delta$
    $\tilde{\mathcal{A}} \longleftarrow \mathcal{A}$
    **while** $\Delta > 0$ **do**
        $a = \arg\min_{a\in\tilde{\mathcal{A}}} A^\mathcal{D}(s, a)$
        $\Delta_a = \min\{\Delta, \beta(a|s)\}$
        $\tilde{\mathcal{A}} \longleftarrow \tilde{\mathcal{A}}/a$
        $\Delta \longleftarrow \Delta - \Delta_a$
        $\pi(a|s) \longleftarrow \pi(a|s) - \Delta_a$
---

### C.7 THE PPO OBJECTIVE FUNCTION

Recently, (Schulman et al., 2017) have suggested a new surrogate objective function, termed *Proximal Policy Optimization* (PPO), that heuristically should provide a reliable performance as in TRPO without the complexity of a TRPO implementation:

$$J^{PPO}(\pi) = \mathbb{E}_{s \sim \beta} \left[ \sum_{a \in \mathcal{A}} \beta(a|s) \min \left( A^\beta(s,a) \frac{\pi(a|s)}{\beta(a|s)}, A^\beta(s,a) \operatorname{clip} \left( \frac{\pi(a|s)}{\beta(a|s)}, 1 - \varepsilon, 1 + \varepsilon \right) \right) \right],$$

where $\varepsilon$ is a hyperparameter. PPO tries to ground the probability ratio by penalizing negative advantage actions with probability ratios above $1 - \varepsilon$. In addition, it clips the objective for probability ratios above $1 + \varepsilon$ so there is no incentive to move the probability ratio outside the interval $[1 - \varepsilon, 1 + \varepsilon]$. However, we show in the following that the solution of PPO is not unique and is dependent on the initial conditions, parametric form and the specific optimization implementation. This was also experimentally found in (Henderson et al., 2017). The effect of all of these factors on the search result is hard to design or predict. Moreover, some solutions may have unbounded probability ratios, in this sense, $J^{PPO}$ is not safe.

First, notice that PPO maximization can be achieved by ad hoc maximizing each state since for each state the objective argument is independent and there are no additional constraints. Now, for state $s$, let's divide $\mathcal{A}$ into two sets: the set of positive advantage actions, denoted $\mathcal{A}^+$, and the set of negative advantage actions, $\mathcal{A}^-$. For convenience, denote $c_i = \frac{\pi(a_i|s)}{\beta(a_i|s)}$ and $\beta_i = \beta(a_i|s)$. Then, we can write the PPO objective of state $s$ as

$$J^{PPO}(\pi, s) = \sum_{a_i \in \mathcal{A}^+} \beta_i A_i^\beta \min (c_i, 1 + \varepsilon) - \sum_{a_i \in \mathcal{A}^-} \beta_i (-A_i^\beta) \max (c_i, 1 - \varepsilon).$$

Clearly maximization is possible (yet, still not unique) when setting all $c_i = 0$ for $a_i \in \mathcal{A}^-$, namely, discarding negative advantage actions. This translates into a reroute maximization with parameters $(c_{\min}, c_{\max}) = (0, 1 + \varepsilon)$

$$\arg\max_{c_i}[J^{PPO}(\pi, s)] = \arg\max_{c_i} \left[ \sum_{a_i \in \mathcal{A}^+} \beta_i A_i^\beta \min (c_i, 1 + \varepsilon) \right] = \arg\max_{c_i} \left[ \sum_{a_i \in \mathcal{A}^+} c_i \beta_i A_i^\beta \right]$$

for $c_i \leq 1 + \varepsilon$. The only difference is that the sum $\sum_i c_i \beta_i = 1 - \Delta$ may be less then 1. In this case, let us take the unsafe approach and dispense $\Delta$ to the highest ranked advantage. It is clear that partition of $\Delta$ is not unique, and even negative advantage actions may receive part of it as long as their total probability is less than $(1-\varepsilon)\beta_i$. We summarize this procedure in the following algorithm.

---

**Algorithm 3:** Ad hoc PPO Maximization

**Data:** $s$, $\beta$, $A^\beta$, $\varepsilon$
**Result:** $\{\pi(a|s),\ a \in \mathcal{A}\}$
**begin**

    $\tilde{A} = \{\tilde{A}^+, \tilde{A}^-\}$
    $\tilde{\mathcal{A}} \longleftarrow \mathcal{A}^+$
    $\Delta \longleftarrow 1$
    $\pi(a|s) \longleftarrow 0\ \forall a \in \mathcal{A}$
    **while** $\Delta > 0$ *and* $|\tilde{\mathcal{A}}| > 0$ **do**

        $a = \arg\max_{a \in \tilde{\mathcal{A}}} A^\beta(s, a)$
        $\Delta_a = \min\{\Delta, (1 + \varepsilon)\beta(a|s)\}$
        $\tilde{\mathcal{A}} \longleftarrow \tilde{\mathcal{A}}/a$
        $\Delta \longleftarrow \Delta - \Delta_a$
        $\pi(a|s) \longleftarrow \pi(a|s) + \Delta_a$

    $a = \arg\max_{a \in \mathcal{A}} A^\beta(s, a)$
    $\pi(a|s) \longleftarrow \pi(a|s) + \Delta$

---

## D  TAXI EXPERIMENT: TECHNICAL DETAILS

In the paper, we presented three selection types of the subset of the optimal policy, i.e. $\mathcal{S}_i^*$: (1) row selection (2) column selection and; (3) random selection. In the Taxi environment the states are enumerated $\{s_j\}_{j=1}^{500}$. For $N$ players, the definitions for our the selection types are
**Row Selection:**

$$\left\{ s_j \in \mathcal{S}_i^* \middle| \left( j - i\frac{500}{N} \right) (\mathrm{mod}\,500) < 250 \right\}$$

**Column Selection:**

$$\left\{ s_j \in \mathcal{S}_i^* \middle| (j + i)\,(\mathrm{mod}\,N) < \frac{N}{2} \right\}$$

**Random Selection:** Randomly (and uniformly) choose 250 states out of the 500 state for each different $\mathcal{S}_i^*$.

For the improvement step experiment, we evaluated the behavior policy $\beta$ with a tabular representation, i.e., $\beta(a|s) = \frac{N_{s,a}}{N_s}$ (see the first item in Definition 4.1). For Monte-Carlo learning, we applies the following update rule

$$Q^\beta(s,a) \longleftarrow Q^\beta(s,a) + \alpha \left( R - Q^\beta(s,a) \right), \tag{13}$$

where $\alpha = 0.1$ is a learning rate and $R_{s,a}(j)$ is a single instance of the total discounted return from a state-action pair $s, a$, up to the terminal state. For Temporal Difference learning, we applied

$$Q^\beta(s,a) \longleftarrow Q^\beta(s,a) + \alpha \left( r + \gamma \sum_{a'} \beta(a'|s')Q^\beta(s',a') - Q^\beta(s,a) \right), \tag{14}$$

with the same learning rate. Notice that in this experiment we did not evaluated $Q^\beta$ from a $V^\beta$ evaluation since for the tabular case, $\beta$ is perfectly calculated with $\beta(a|s) = \frac{N_{s,a}}{N_s}$.

## E  ATARI EXPERIMENT: TECHNICAL DETAILS

### E.1  DATASET PREPROCESSING

The dataset (Kurin et al., 2017) (Version 2) does not contain an end-of-life signal. Therefore, we tracked the changes of the life icons in order to reconstruct the end-of-life signal. The only problem with this approach was in Qbert where the last life period has no apparent icon but we could not match a blank screen since, during the episode, life icons are flashed on and off. Thus, the last end-of-life signal in Qbert is missing. Further, the dataset contained some defective episodes in which the screen frames did not match the trajectories. We found them by comparing the last score in the trajectory file to the score that appears in the last or near last frame.

We also found some discrepancies between the Javatari simulator used to collect the dataset and the Stella simulator used by the OpenAI gym package where we evaluated the agents' scores:

- The Javatari reward signal has an offset of $-2$ frames. We corrected this shift in the pre-processing phase.

- The Javatari actions signal has an offset of $\sim -2$ frames (depending on the action type), where it is sometimes recorded non-deterministically s.t. the character executed an action in a frame before the action appeared in the trajectory file. We corrected this shift in the preprocessing phase.

- The Javatari simulator is not deterministic, while Stella can be defined as deterministic. This has the effect that icons and characters move in different $mod_4$ order, which is crucial when learning with frame skipping of 4. Thus, we evaluated the best offset (in terms of score) for each game and sampled frames according to this offset.

- There is a minor difference in colors/hues and objects location between the two simulators.

### E.2    LEARNING FROM HUMAN PLAYERS

The Atari dataset introduced a further challenge of learning from *human players*. Contrary to RL agents, while much of the developed theory in previous sections assumes observation of *Markovian* policies, humans do not play with MR policies. We found that two significant sources of Non-Markovian behavior are: (1) delayed response and (2) action repetition. The first happens due to an unavoidable delay between eye and hand movement. In simple words, humans respond to a past state. The second implies that actions depend on action history, which violates the Markovian assumption. Practically, when using a NN classifier, such Non-Markovian behavior hurts the classification since the network learns to predict past actions. We found that training the network to predict an action in the near future (6 frames away, i.e. 0.1 seconds) can mitigate such a non-Markovian nature. This way there is less correlation between the present state and the executed action, and the human delayed response is mitigated.

### E.3    NETWORK ARCHITECTURE

A finite dataset introduces overfitting issues. To avoid overfitting, DQN constantly updates a replay buffer with new samples. In a finite dataset this is not possible, but, contrary to many Deep Learning tasks, partitioning into training and validation sets is also problematic: random partitions introduce a high correlation between training and testing, and blocking partitioning (Racine, 2000) might miss capturing parts of the action-states space. Moreover, the ultimate learning goal, i.e. the playing score, is not necessarily captured via the loss function score. In LfO, evaluating the agent's score to avoid overfitting violates the assumption of learning only from the dataset. Fortunately, we found that Dropout (Srivastava et al., 2014), as a source of regularization, improves the network's resiliency to overfitting. In addition, it has the benefit of better generalization to new unseen states since the network learns to classify based on a partial set of features. We added two sources of dropout: (1) in the input layer (25%) and (2) on the latent features i.e. before the last layer (50%).

We also found that for a finite dataset with a constant statistics, Batch Normalization (BN) can increase the learning rate. Therefore, we used BN layer before the ReLu nonlinearities. Note that for an online RL algorithm, BN may sometimes impede learning because the statistics may change during the learning process.

To estimate the advantage for the PPO objective, we used a trick inspired by the Duelling DQN architecture (Wang et al., 2015). We added a single additional output to the last layer of the $Q$ network which represents the value $V^\beta$. The other $|\mathcal{A}|$ outputs represents the unnormalized advantages $\{\tilde{A}_i\}_i$ and the $Q$-function outputs are therefore expressed as

$$Q_i^\beta = V^\beta + \tilde{A}_i - \sum_j \tilde{A}_j \beta_j, \tag{15}$$

where $\{\beta_j\}_j$ are the outputs of the policy network. Note that $\sum_i Q_i^\beta = V^\beta$. The normalized advantage is therefore $A_i^\beta = \tilde{A}_i - \sum_j \tilde{A}_j \beta_j$.

Finally, we also shaped a reward for an end-of-life signal (for all our experiments including DQfD). Usually, DQN variants set an end-of-life signal only as a termination signal so that the agent learns that near end-of-life states have a 0 value and, as a consequence, states that dodge termination are preferred. Since RBI policy is stochastic and does not just choose the best one, it is helpful to differentiate between near zero reward and termination. Thus, we added a negative reward (-1) for a terminal state.

To summarize, we used two DQN style networks: one for $\beta$ and the second for $Q^\beta$. We used the same network architecture as in (Mnih et al., 2015) except for the following add-ons:

- A batch normalization layer before nonlinearities.

- A 25% Dropout in the input layer.

- A 50% Dropout in the latent features layer. To compensate for the dropped features we expanded the latent layer size to 1024, instead of 512 as in the original DQN architecture.

- An additional output that represents the state's value.

The overall architecture is depicted in figure 5. For the behavioral cloning, we used the Cross-Entropy Loss. For the value network, though the theoretical bound in the paper is calculated for the $L_1$ loss, we found that slightly better results are obtained with the $MSE$ loss. This may be due to better regression of outliers. All results reported in the paper used the MSE loss for value.

### E.4 LEARNING PROCESS

A single iteration of our learning process is depicted with the following PyTorch style pseudo-code. Value and Behavioral networks are trained simultaneously on the same sample. The state is passed through the beta net once in training mode (with dropout) and once in evaluation mode (without dropout) for the advantage calculation.

```
beta_net.train()
value_net.train()

for sample in train_loader:

    s, a, r_mc = sample
    beta = beta_net(s)
    beta_net.eval()
    beta_eval = beta_net(s)
    beta_net.train()
    q, adv, v = value_net(s, beta_eval)
    q_a = q[a]
    loss_v = MSELoss(q_a, r_mc)
    loss_beta = CrossEntropyLoss(beta, a)

    # execute gradient descent with Adam
    # optimization over loss_beta and loss_v
```

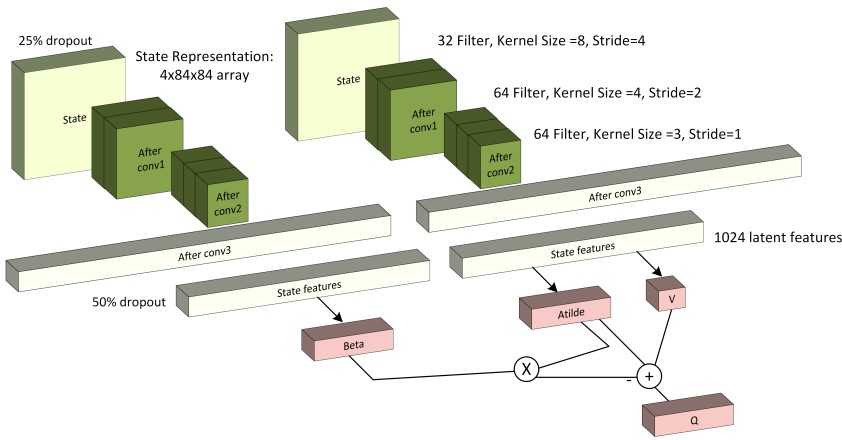

Figure 5: Network architecture

## E.5 EVALUATION

The execution of a RBI policy at evaluation time is depicted with the following pseudo-code.

```
beta_net.eval()
value_net.eval()

t = 0
score = 0
s = env.reset()

while not t:

    beta = beta_net(s)
    q, adv, v = value_net(s, beta)
    pi = max_reroute(beta, adv, c_min, c_max)
    a = choice(pi)

    s, r, t = env.step(a)

    score += r
```

## E.6 Hyperparameters tables

Table 1: Policy and $Q$-value networks Hyperparameters

| Name | Value |
|---|---|
| Last linear layer size | 1024 |
| Optimizer | Adam |
| Learning Rate | 0.00025 |
| Dropout [first layer] | 0.25 |
| Dropout [last layer] | 0.5 |
| Minibatch | 32 |
| Iterations | 1562500 |
| Frame skip | 4 |
| Reward clip | -1,1 |

Table 2: DQfD Hyperparameters

| Name | Value |
|---|---|
| Optimizer | Adam |
| Learning Rate | 0.0000625 |
| $n$-steps | 10 |
| Target update period | 16384 |
| $l_{a_E}$ | 0.8 |
| Priority Replay exponent | 0.4 |
| Priority Replay IS exponent | 0.6 |
| Priority Replay constant | 0.001 |
| Other parameters | As in policy/value networks (except Dropout) |

## F  Final Policy Performance Table

Table 3: Final scores table

| Method | MsPacman | Qbert | Revenge | SpaceInvaders |
|---|---|---|---|---|
| Humans | 3024 | 3401 | 1436 | 634 |
| Behavioral cloning | 1519 | 8562 | 1761 | 678 |
| Reroute-$(0.5, 1.5)$ | 1550 | 12123 | 2379 | **709** |
| Reroute-$(0.25, 1.75)$ | **1638** | 13254 | **2431** | 682 |
| Reroute-$(0, 2)$ | 1565 | **14494** | 473 | 675 |
| TV$(0.25)$ | 1352 | 5089 | 390 | 573 |
| PPO$(0.5)$ | 1528 | 14089 | 388 | 547 |
| DQfD | 83 | 1404 | 1315 | 402 |

