# OpenReview forum: "Safe Policy Learning from Observations"
_ICLR.cc/2019/Conference_

### Official Review · AnonReviewer3 · 2018-11-02
**Assumption in this paper significantly deteriorate the significance of the results**

**Rating:** 5
**Confidence:** 4

**Review:**

In this paper, the authors study the problem if learning for observation, a reinforcement learning setting where an agent is given a data set of experiences from a potentially arbitrary number of demonstrators. The authors propose a method which deploys these experience to initialize a place. Then estimate the value of this policy in order to improve it.

The paper is well written and it is easy to follow.

Most of the theoretical results are interesting and the derivations are kinda straightforward but not fully matching the main claim in the paper. Mainly the contribution in this paper heavily depends on an assumption that Q^D and Q^\beta are close to each other. This assumption simplifies the many things resulting in a simple algorithm. But this assumption is too strong while the main challenge in the line of learning from observation comes from the fact that this assumption does not hold. Under this assumption and the similarity in distributions mentioned in proposition 4.2 make the contribution of this paper significantly weak.

Please let me know if you do not actually use this assumption in your results and justification.

---

> ### Author Response · Authors · 2018-11-10
> **Response to Reviewer 3**
>
> We thank the reviewer for his review.
> We would appreciate if the reviewer could reevaluate the significance of our work in the light of the following answer:
>
>
> Please let us focus in the first part of our answer in section 5:
>
> The entire derivation in section 5 does not necessarily assume that Q^D equals Q^\beta. Our derivation only assumes that we are given some approximation of Q and required to take an improvement step. In this sense, Lemma 5.2 is applicable also for a standard setup where only a single policy is observed. When a single policy is observed, the entire error -i.e. \eps(s,a) in Eq. (7) is due to the randomness in the sampled trajectories. In this case, if Q was learned with MC trajectories the SE of \eps is given by Eq. 6.
> (Please refer to our answer to reviewer 1 regarding a similar justification for TD learning.)
>
> In the multiple policies setup, we may decompose the estimation error into two terms \eps(s,a) = \eps^D(s,a) + \eps'(s,a) where \eps^D is the fixed error due to the difference between Q^D and Q^\beta and \eps' is the r.v. error of estimating Q^D from N MC trajectories.
> Notice that the number of visitations is still N_s * beta(a|s) thus the SE of \eps' is still given by Eq. 6 and hence, the need for the reroute constraint remains.
> While \eps^D(s,a) may contribute to the improvement penalty, it is bounded by the proximity of P(pi|s) and P(pi|\tilde{s},\tilde{a}\xrightarrow[k]{} s). All our experiments suggest that, generally, the improvement due to taking favorable actions exceeds possible degradation due to minor evaluation errors. This pattern repeats both in the tabular and in the deep cases. However, the second term \eps'(s,a) may still grow to the extreme if uncontrolled and this is the main source of performance degradation when taking greedy/TV constrained steps or a single PPO optimization.
>
> We would also like to highlight two important results which we believe are important to the RL literature.
>
> First, our derivation shows that both TRPO and PPO overlooked this consideration in their trust region definition. Therefore while they are considered as a guaranteed improvement procedure, this caveat may lead to potential drops in their learning pattern. The reroute constraint is designed to limit such drops.
>
> In addition, our method is resilience to small errors of \eps^D(s,a) such that if action ranking does not changes, then improvement is guaranteed. This is in contrast to policy gradient methods which relies on the quantitative Q-value and not the categorical action ranking. Lastly, since we do not optimize the policy network, our improvement step is exact and free of neural networks optimization pitfalls such as convergence to local minima or overfitting. This is true for a 1-step improvement and in an iterative improvement process it is naturally generalized to something very similar to the "supervised policy update" paper [1] but contrary to their constraints our constraint is well motivated. Please also refer to our answer to reviewer 1 for potential generalization to iterative RL.
>
>
> Regarding section 4:
>
> All reviewers raised questions about the seemingly unpopular choice of learning from MC returns. Particularity since directly estimating \beta with Off-Policy methods is justified and does not require any assumptions or approximations about the difference between Q^D and Q^\beta. Our experiments with 1-step TD learning of Q^\beta provided lower scores than with MC evaluation. This led us to explain the rationale of MC learning via propositions 4.1, 4.2. In order to properly compare between TD and MC methods, we will make the following amends:
>
> (1) Present the two approaches of evaluating Q^\beta: 1-step TD and approximated MC.
> (2) add our results of RBI learning with the evaluated 1-step TD Q^\beta.
> (3) discuss possible reasons why MC is better in this setting.
>
> We postulate that there are 2 main reasons why TD is inferior to MC in finite small datasets:
> (1) TD require evaluation of \beta, even with 1-step since Q^\beta(s,a) = r(s,a) + \gamma * \sum_{a'} \beta(a'|s')Q^\beta(s',a'). Since our evaluation of \beta inherently contains errors, they propagate to the Q-value evaluation.
> (2) TD, in contrast to MC, contains bootstrapping. Therefore, errors from poorly evaluated states propagate to other states. In an iterative RL settings, these can be corrected but in a fixed size dataset it is better to avoid bootstrapping with MC methods.
>
>
>
> References:
>
> [1] Vuong QH, Zhang Y, Ross KW. Supervised Policy Update. arXiv preprint arXiv:1805.11706. 2018 May 29.

---

### Official Review · AnonReviewer2 · 2018-11-02
**Contribution to safe RL with a weak empirical validation**

**Rating:** 5
**Confidence:** 3

**Review:**

The paper plugs the ideas of TRPO/PPO into the value based RL. Though there is no big surprise in terms of the tools used, this is interesting to know that safe policy improvement is possible in this setting.

Nevertheless for a conference as ICLR which is interested in the performance of ML tools, I have two concerns:

- The scores obtained on all tests tasks on Atari game are quite far from the state of the art. As an example OpenAI announced to be able to score 74k at Montezuma's revenge with a single demonstration using PPO and a carefull selection of the initializations states (see  blog post https://blog.openai.com/learning-montezumas-revenge-from-a-single-demonstration/). I understand that the setting is not directly comparable but the goal of RL is to learn good policies. This remark would vanish is the authors could come with a real use case where for some reason their approach is the best performer.

- The proposed approach is benchmarked wrt few algorithms while there exist a lot in the safe RL literature. The setting is often slightly different but adaptation is often possible. In particular I'd like more positioning wrt what is proposed by the work of Petrik&all (https://papers.nips.cc/paper/6294-safe-policy-improvement-by-minimizing-robust-baseline-regret.pdf the paper is cited but the first author is incorrect). What are the deep differences that make this paper setting more interesting (in terms of what can be done from an applied perspective) or more challenging in terms of mathematical tools. Here I feel the core difference is a comparison against an average of policies which becomes the new baseline to beat.

Also not that at EWRL'18 an alternative approach for value based safe RL was presented https://arxiv.org/pdf/1712.06924.pdf

---

> ### Author Response · Authors · 2018-11-10
> **Response to Reviewer 2**
>
> We thank the reviewer for the insightful comments.
>
> Regarding other Atari LfD works like [1,2] and "OpenAI - Learning Montezuma’s Revenge from a Single Demonstration":
>
> These works utilize the demonstrations to guide a RL agent which actively interacts with the environment (first two via regularization and the last via curriculum learning). In contrast, our reported score is obtained without any learning from self interaction with the environments. Instead we focus on finding safe initial policy which beats the average behavior.
>
> A fairer comparison would have been with the initial score reported by DQfD (which is the score after a training phase of 750,00 mini-batches only with the demonstration data). In this case, while they do not provide the exact score, their starting point in the graph shows that their initial policy (while much better than random - as expected) is inferior to our learned policy (mspacman < 1000, revenge ~1300, qbert ~5000, spaceinvaders is absent). However, this comparison is still a little bit misleading since the datasets of demonstrations is different and while their dataset is of few demonstrations of a single expert we have a dataset much more observations of many different players (the statistics of the scores of those players is found at [5]
> ). Therefore, in order to test our approach against this benchmark, we implemented DQfD and trained it on our dataset. The score are provided in our paper and show that it obtains lower score than our behavioral cloning phase in all games. On the other hand, RBI significantly improves upon the behavioral cloning  in 2 out 4 games (Qbert +83% and Revenge +36%) and slightly improves Mspacman (+4.5%) (in spaceinvaders we observe -2% drop).
>
> In contrast to this line of work, both papers [3,4] consider a model based approach where they assume enough historical data to build a model and a simulator. Specifically [3] assumes that it has an estimated model with a bounded error on the transition probabilities. Moreover, its approximated solution requires "fixing the model" by counting over the entire state-space. These assumptions are often too restrictive  when we learn a policy from raw sensory input with Deep Neural Network and a fixed size dataset. Therefore, we resort to a model-free approach.
>
> To conclude, as shown by our experimental results, our approach is the best performer when you are required to provide a safe initial policy given multiple weak demonstrators in learning from raw sensory input. This setup is of practical interest in the autonomous car setting for example where you can collect human drivers statistics but you must provide initial safe policy. Another example is Robot teleoperation where a robot is teleoperated by many operators (with different policies) for a limited period and later the robot is required to act autonomously.
>
> Please also refer to our answer to reviewer 3, regarding learning the Q-value from MC vs TD method.
>
>
> References:
>
> [1] Hester T, Vecerik M, Pietquin O, Lanctot M, Schaul T, Piot B, Horgan D, Quan J, Sendonaris A, Dulac-Arnold G, Osband I. Deep Q-learning from Demonstrations. arXiv preprint arXiv:1704.03732. 2017 Apr 12.
> [2] Pohlen T, Piot B, Hester T, Azar MG, Horgan D, Budden D, Barth-Maron G, van Hasselt H, Quan J, Večerík M, Hessel M. Observe and Look Further: Achieving Consistent Performance on Atari. arXiv preprint arXiv:1805.11593. 2018 May 29.
> [3] Ghavamzadeh M, Petrik M, Chow Y. Safe policy improvement by minimizing robust baseline regret. InAdvances in Neural Information Processing Systems 2016 (pp. 2298-2306).
> [4] Laroche R, Trichelair P, Asri LE. Safe Policy Improvement with Baseline Bootstrapping. arXiv preprint arXiv:1712.06924. 2017 Dec 19.
> [5] Kurin V, Nowozin S, Hofmann K, Beyer L, Leibe B. The Atari Grand Challenge Dataset. arXiv preprint arXiv:1705.10998. 2017 May 31.

---

### Official Review · AnonReviewer1 · 2018-11-04
**A good paper with interesting theory and algorithmic contribution. The weaknesses are the clarity as well as the limited experiments.**

**Rating:** 5
**Confidence:** 4

**Review:**

The paper looks at learning a policy from multiple demonstrators which should also be safely improved by an reinforcement learning signal. They define the policy as a mixture of policies from the single demonstrators. The paper gives a new way to estimate the value function of each policy where the overall policy is defined as mixture of the single policies. The paper subsequently looks at the standard error of the value function estimation and then define the policy improvement step in the presence of value estimation error. The resulting reroute constraint for the policy improvement step is evaluated on the taxi toy task as well as on 4 different atari domains.

This paper presents an interesting ideas which is also based on an exhaustive theoretical derivation. However, the paper is lacking clarity and motivation which makes it almost impossible to understand at the first pass. Moreover, the presented results are promising but not exhaustive and the resulting algorithm is also restricted to discrete action domains. More comments see below:

- The paper consists of 2 parts, the average behavior policy and its value function and the safe policy improvement step. The relation between these two parts are not clear. Is the policy improvement step only working if the policy is defined as in section 4 and the value function computed as in section 4?
- Proposition 4.2 needs to be much better motivated and explained. It is totally unclear at this part of the paper why proposition 4.2 is used.
- Please explain why proposition 4.2 indicates that Q^D \approx Q^\beta
- The selection type of S in the taxi example is also unclear.
- How would we solve Equation 8 with continuous actions / parametrized policies \pi? Without this extension, the algorithm is quite restricted.
- the figure captions need to be much more exhaustive. I am not sure I understand the x axis of Figure 4 (right). What iterations are shown here? We only do one improvement step of the behavior policy, without any resembling, is that right?
- Could we also use a similar policy update for policy improvement in reinforcement learning?
- Could you add an algorithm box for estimating the Q-function? Do we estimate every Q-function in isolation using MC estimates and then just use the weighted average?
- It would be interesting to also compare the value function learning method proposed in the paper in isolation to other value function learning methods such as DQN. while the presented method is simple (learn from MC estimates), this is also known to be very data inefficient.

---

> ### Author Response · Authors · 2018-11-10
> **Response [2/2] to Reviewer 1**
>
> The figure captions need to be much more exhaustive:
>
> Agreed. Somehow we missed this part. Each iteration is a backward learning step applied both to the learned value and the learned policy. During the entire process, we evaluated 3 different policies: (1) behavior (2) behavior + TV constrained step and; (3) behavior + reroute constrained step. In addition, we plot the learning curve of the DQfD baseline. In addition, the left figure plots also the performance of a PPO optimization step applied to the behavior policy learned after ~1.5M iterations. We will update this information to the figures and the comparison with PPO paragraph.
>
>
> Q:
> How would we solve Equation 8 with continuous actions?
>
> A:
> Eq. 8 and the reroute constraint is motivated by the Standard Error of the Q-value evaluation and Lemma 5.2. Generalization to continuous action space requires several considerations. First, for Learning from Observations (LfO), learning the average policy requires some model for the density function of \beta, we assume that unlike common parameterizations of iterative learners, Guassian model or even Mixture of Gaussian would not qualitatively represent the \beta distribution. It may be possible that a quantile network is a proper choice. The second challenge is to quantify or bound the expected error or its variance. Also here, there is no general solution and it depends on the parameterization. To conclude, we do believe that designing a safe trust region for continuous control is a desirable goal and it may provide better results in real-world data than trust regions that do no take into account estimation error (like KL). However, we believe that it deserves an additional future research.
>
>
> (1) Could you add an algorithm box for estimating the Q-function? (2) Do we estimate every Q-function in isolation using MC estimates and then just use the weighted average?
>
> (1)Sure.
> (2) One of the key points in our approach is avoiding estimating the Q-function of each and every different player. This is both computationally prohibitive and in addition, it requires estimating the conditional probability P(p^i|s), i.e. the probability of sampling a player p^i given a sampled state s. We show that learning with MC trajectories and L1-loss circumvents this obstacle.
>
>
> About the comparison to DQN:
>
> Notice that we indeed compared our results to DQfD [1] which is the DQN method with additional 2 regularization terms designated to a learning from demonstrations trajectories: L2-regularization to prevent overfitting a fixed small dataset and more importantly a penalty for choosing actions different from the action demonstrated by the demonstrator. For all the games in our dataset, we show that our method is always better than DQfD. In our experiments, DQN (i.e. DQfD without the regularization terms) provided lower scores than DQfD. Please see also the comment to reviewer 2 regarding the correct comparison between DQfD and our RBI method.
>
> Regarding learning the Q-value from MC vs TD method, please refer to our answer to reviewer 3.
>
>
> References:
>
> [1] Hester T, Vecerik M, Pietquin O, Lanctot M, Schaul T, Piot B, Horgan D, Quan J, Sendonaris A, Dulac-Arnold G, Osband I. Deep Q-learning from Demonstrations. arXiv preprint arXiv:1704.03732. 2017 Apr 12.
> [2] Kearns MJ, Singh SP. Bias-Variance Error Bounds for Temporal Difference Updates. InCOLT 2000 Jun 28 (pp. 142-147).

---

> ### Author Response · Authors · 2018-11-10
> **Response [1/2] to Reviewer 1**
>
> We thank the reviewer for the insightful comments.
>
> Q:
> Is the policy improvement step only working if the policy is defined as in section 4 and the value function computed as in section 4?
>
> A:
> Generally, reroute constrained policy improvement step is applicable and justifiable with other setups including iterative learning both with MC and TD methods. In this work we focused on MC returns since the theoretical motivation for this step is easily explained in the tabular setting with MC returns and this also coincides with our method for learning the Q-value. For TD methods, it is worth to consider the upper bound of the value evaluation error, this was shown to be proportional to  1 / \sqrt{N} where N is the number of visitations [2]. This is equivalent to the 1/\sqrt{N} term appears in the standard deviation of the error which is analyzed in our work. Therefore, the same argument for the reroute constraint holds also for TD-learning.
>
> This leads to your question about iterative RL,
> Q:
> Could we also use a similar policy update for policy improvement in reinforcement learning?
>
> A:
> Yes. A direct consequence of this work is a new iterative policy algorithm which has two parts:
> (1) the learner estimates the past policy (with KL divergence loss) and evaluates its Q-value function.
> (2) the actor takes the policy and its value and calculates in each step a policy which maximizes the reroute constraint (and then saves to a memory buffer the generated policy, the state and the reward). We identify two important advantages of this approach:
> (1) With respect to greedy algorithms like DQN: it allows to increase the safety level of each improvement step - as we showed in the paper, greedy is a very precarious approach particularly in less explored areas.
> (2) With respect to policy gradient methods: The optimization of the policy, i.e. the calculation \beta \to \pi does not depend on the parametric form and hence it is exact and avoids NN optimization traps (overfitting, local minima etc). Therefore, the rerouted policy (behavioral followed by Max-Reroute optimization) should provide better trajectories and support faster learning rate.
> Note that to substantiate these claims for iterative RL an additional experimental work is required, therefore we provide them here just as a hunch and motivation. However, in this work we believe that we have firmly proved them (both theoretically and experimentally) for a single improvement step: our Reroute constraint is better than greedy step/ TV constraint step and a single PPO optimization step.
>
>
> About proposition 4.2:
>
> Proposition 4.2 suggests a hypothetical policy which has a value function of Q^D. The structure of this policy, termed \beta^D_{\tilde{s},\tilde{a}} is similar to the structure of the average behavior \beta with a single difference: the weights P(p^i|s) in \beta are changed to P(p^i|\tilde{s},\tilde{a} \xrightarrow[k]{} s). This is equivalent to P(p^i|s) but with a dataset \mathcal{D}_{\tilde{s},\tilde{a}}^k which is a subset of D and contains all the state-action pairs that are k-step away from a state-action pair \tilde{s},\tilde{a}. Therefore, the difference |Q^\beta - Q^D| is bounded by how much P(p^i|\tilde{s},\tilde{a} \xrightarrow[k]{} s) deviate from P(p^i|s). We experimentally show that: (1) this deviation is generally small - i.e. the TV distance between \beta and \beta^D_{\tilde{s},\tilde{a}} is low (2) if we only consider the Q-value ranking (since our improvement step is based only on the action ranking), we obtain very high Pearson's rank correlation score. In addition, please refer to our answer to reviewer 3 concerning: (1) why we claim that Q^D is sufficient for our policy improvement step and; (2) comparison to TD methods.
>
>
> About the selection types in the taxi example:
>
> To compare between Q^D and Q^\beta we tried many datasets with different synthetic policies which were based on a different mixture between semi-random actions (taking a random action in 75% and optimal policy otherwise) and optimal actions. The mixture was based on different states allocations (i.e. selection) wherein one set, named S* the policy is optimal and in the complement the policy is semi-random. The exact definitions of selections are in appendix D. Essentially, they simulate different types of datasets:
> (1) random selection simulates a dataset of weak demonstrators with different policies.
> (2) the two other selections simulate datasets where different players demonstrate optimal policies in different parts of the MDP (those parts are unknown to the agent).

---

### Author Response · Authors · 2018-11-27
**Summary of the revision**

Dear reviewers,

Thank you for your thoughtful feedback. We have updated the manuscript according to your suggestions:

1. In section 4 (Average Behavior and Its Value Function) we introduced two alternatives for estimating the value function with their pros and cons:
(1) TD learning - without using the average behavior estimation.
(2) Approximated MC learning.

2. In the Taxi example (section 5 - Safe Policy Improvement) we compared the improvement steps with TD and MC learning.

3. We reran the Atari experiments with both TD and MC estimators - results are presented and discussed in section 6.

To summarize, our experiments show that:
(1) In the tabular case (Taxi example), there is no clear winner between TD and MC.
(2) As we expected, with NN (Atari example) the MC learning method provided much better results.

---

### Meta-Review · Area_Chair1 · 2018-12-14
**Interesting idea, but limited applicability**

**Confidence:** 4
**Recommendation:** Reject

**Metareview:**

The paper studies safer policy improvement based on non-expert demonstrations.  The paper contains some interesting ideas, and is supported by reasonable empirical evidence.  Overall, the work has a good potential.  The author response was also helpful.  That said, after considering the paper and rebuttal, the reviewers were not convinced the paper is ready for publication, as the significance of this work is limited by a rather strong assumption (see reviews for details).  Furthermore, the presentation of the paper also requires some work to improve (see reviews for detailed comments).